# Sustained store-operated calcium entry utilizing activated chromatin state leads to instability in iTregs

Huiyun Lyu[1,2†], Guohua Yuan[3,4†], Xinyi Liu[1,5], Xiaobo Wang[1,5], Shuang Geng[6], Tie Xia[1,5], Xuyu Zhou[7,8], Yinqing Li[3,4], Xiaoyu Hu[1,2,5], Yan Shi[1,2,5,6]*

[1]Institute for Immunology, Beijing Key Lab for Immunological Research on Chronic Diseases, Tsinghua University, Beijing, China; [2]Tsinghua-Peking Center for Life Sciences, Tsinghua University, Beijing, China; [3]IDG/McGovern Institute for Brain Research and School of Pharmaceutical Sciences, Tsinghua University, Beijing, China; [4]MOE Key Laboratory of Bioinformatics, Center for Synthetic and Systems Biology, School of Pharmaceutical Sciences, Tsinghua University, Beijing, China; [5]Department of Basic Medical Sciences, School of Medicine, Tsinghua University, Beijing, China; [6]Department of Microbiology, Immunology and Infectious Diseases, Snyder Institute, University of Calgary, Calgary, Canada; [7]Key Laboratory of Pathogenic Microbiology and Immunology, Institute of Microbiology, Chinese Academy of Sciences, Beijing, China; [8]University of Chinese Academy of Sciences, Beijing, China

*For correspondence: yanshiemail@mail.tsinghua.edu.cn

†These authors contributed equally to this work

**Abstract** Thymus-originated tTregs and in vitro induced iTregs are subsets of regulatory T cells. While they share the capacity of immune suppression, their stabilities are different, with iTregs losing their phenotype upon stimulation or under inflammatory milieu. Epigenetic differences, particularly methylation state of *Foxp3* CNS2 region, provide an explanation for this shift. Whether additional regulations, including cellular signaling, could directly lead phenotypical instability requires further analysis. Here, we show that upon TCR (T cell receptor) triggering, SOCE (store-operated calcium entry) and NFAT (nuclear factor of activated T cells) nuclear translocation are blunted in tTregs, yet fully operational in iTregs, similar to Tconvs. On the other hand, tTregs show minimal changes in their chromatin accessibility upon activation, in contrast to iTregs that demonstrate an activated chromatin state with highly accessible T cell activation and inflammation related genes. Assisted by several cofactors, NFAT driven by strong SOCE signaling in iTregs preferentially binds to primed-opened T helper ($T_H$) genes, resulting in their activation normally observed only in Tconv activation, ultimately leads to instability. Conversely, suppression of SOCE in iTregs can partially rescue their phenotype. Thus, our study adds two new layers, cellular signaling and chromatin accessibility, of understanding in Treg stability, and may provide a path for better clinical applications of Treg cell therapy.

## eLife assessment

This study presents the **valuable** finding that sustained calcium signaling in induced-Treg (iTreg) cells can lead to the loss of Foxp3 expression and iTreg identity by altering the chromatin landscape. The evidence supporting the claims of the authors is **convincing**. The work will be of interest to immunologists working on Treg cell therapy.

## Introduction

The regulatory T cells (Tregs) are key to maintain peripheral immune tolerance and to block excessive inflammatory reactions. There are two main populations of Tregs. The prototypical one originates from the thymus (tTreg), sharing a developmental program similar to conventional T cells (*Rudensky, 2011*; *Sakaguchi et al., 2008*). Their TCRs are believed to have higher basal affinity to MHC (major histo-compatibility antigen), and express Foxp3 during CD4 single positive period (*Fontenot et al., 2005*). The other population originates from the periphery (pTregs) via conversion of Tconvs, arising during inflammation with proper antigenic stimulation and inductive cytokine environment (*Curotto de Lafaille et al., 2004*; *Apostolou and von Boehmer, 2004*; *Cobbold et al., 2004*). iTregs, mimicking the induction of pTregs, is a type of Tregs induced in vitro in the presence of TGF-β (tumor growth factor ) (*Chen et al., 2003*; *Fantini et al., 2004*). While iTregs share similarity in suppressive capacities to Tregs in vivo (*Shevach and Thornton, 2014*; *Fu et al., 2004*; *Singh et al., 2009*), they are different from tTregs in several aspects, particularly regarding their stabilities. Although tTregs in some occasions show certain levels of plasticity, their phenotypes are mostly stable, both in vivo and in vitro (*Rubtsov et al., 2010*). iTregs induced in vitro are more labile with marked loss of Foxp3 expression after activation, limiting the duration of their suppression (*Floess et al., 2007*; *Chen et al., 2011*). The short life of iTregs is also a roadblocker of the clinical use of Treg cell therapy, as they cannot be used as an alternative source to replace numerically limited tTregs (*Koenecke et al., 2009*); and in cases where iTregs are raised with targeted specificities, they carry concerns of potential self-reactivity following the loss of Foxp3.

The stability of tTregs is established prior to their exit from the thymus (*Ohkura et al., 2013*). In the thymus, Foxp3 works with other transcription factors, such as NFAT and c-Rel to maintain a steady transcriptional program (*Fu et al., 2012*; *Hill et al., 2007*; *Samstein et al., 2012*). At the same time, the extensive demethylation of *Foxp3* CNS2 is also critical to their stable phenotype, an event likely taking place before Foxp3 expression (*Samstein et al., 2012*; *Ohkura et al., 2012*). While epigenetic differences provide a possible explanation of Treg stability, it is more probable that multiple factors collectively contribute to the difference. Unlike Tconvs that undergo expansion, attrition, and memory states after activation, tTregs lose signature changes and return to their original state following activation (*van der Veeken et al., 2016*). This persistence in phenotype of tTregs is certainly beneficial to the host, yet mechanisms for this maintenance, in contrast to the lack thereof in iTregs are incompletely understood. Previous work suggests that TCR signal as well as ensuing $Ca^{2+}$ activities are weaker in tTregs (*Sumpter et al., 2008*; *Vaeth et al., 2012*; *Yan et al., 2015*), whether the reduced TCR signaling strength is the root of tTreg stability remains an intrigue, particularly strong TCR engagement facilitates iTregs' loss of phenotype.

Here, we report a surprising finding that may contribute to the iTregs' phenotypical change after their activation. We found that ligation of TCR induces long vibration of $Ca^{2+}$ signals in Tconvs that lasts for hours, as typically expected in the store-operated calcium entry (SOCE) activation. Initially similar, this vibration is quickly diminished in tTregs, correlating with their reduced response to TCR stimulation. In contrast, in iTregs, this $Ca^{2+}$ vibration is indistinguishable from Tconvs. The sustained $Ca^{2+}$ signal leads to a strong NFAT activation in iTregs. Regarding chromatin accessibility, with the aid of our initial observation of iTregs from the perspective of activated Tconv, we found iTregs also show an 'activated' chromatin state with T cell activation and inflammation related genes highly open. We demonstrate that the NFAT nuclear translocation driven by sustained calcium signal allows its direct access to prime-opened $T_H$ genes and upregulate their expression, destabilizing iTregs. As expected, blocking CRAC channel reverses the loss of Foxp3 in iTregs in vitro and in vivo. Our work therefore suggests that $Ca^{2+}$ signaling and accessible chromatin state in iTregs are two key causes in their loss of stability following activation, which may be of important value to understand how these two populations of Tregs operate in the disease conditions and lead to better applications of iTregs in the clinic.

## Results

### iTregs use a suppression mechanism similar to that of tTregs

In our previous work, we found that Tregs show strong binding to DCs, which is a key mechanism for blocking DC's ability to engage other Tconvs (*Chen et al., 2017*). The strong binding is due to low basal calcium oscillation of tTregs in the steady state (*Wang et al., 2022*), which leads to diminished

mCalpain activation and commensurate reduced internalization of LFA-1. Furthermore, our previous study showed the low calcium oscillation is maintained by reduced expression of endosomal $Ca^{2+}$ channel Ryr2 by Foxp3 (*Wang et al., 2022*). To ascertain that this regulation is also central to iTregs, we produced iTregs from Tconvs. With plate-bound anti-CD3/CD28 and in the presence of TGF-β, iTregs were induced and the percentage was maximized on day 2. We found that in vitro iTregs show stronger suppression than tTregs on OT-II proliferation in response to epitope peptide loaded DCs (*Figure 1A*). In accordance with the stronger suppression, single-cell force spectroscopy analysis showed that iTregs possessed apparent binding to DCs, even stronger than tTregs (*Figure 1B*). In addition, iTregs also showed increased binding force compared with activated Tconvs (*Figure 1C*), suggesting the strong binding force is a common feature of both types of Tregs, but not Tconvs. We measured the expression of Foxp3 more precisely, and noticed that the protein-level expression of Foxp3 reached its peak on day 2 after the induction (*Figure 1D*). With the accumulation of Foxp3 protein, basal $Ca^{2+}$ oscillation in the treated Tconvs was gradually diminished, with the former leading the latter roughly by 1 day (*Figure 1E*). As expected, the transcription level of Ryr2 was suppressed both in iTregs and tTregs (*Figure 1—figure supplement 1A*), suggesting that the strong binding force with DC may come from reduced calcium oscillation as a result of Foxp3-mediated suppression of Ryr2 transcription. While Ryr2 expression may be regulated by Foxp3, the induction efficiency of Foxp3 was not affected by Ryr2 knock out (*Figure 1—figure supplement 1B*). Therefore, we show iTregs can apply a suppression mechanism similar to tTregs, which operates via the indirect inhibition via DC blockage. We do not know why iTregs show a stronger suppression than tTregs, an event maybe related to their activation in the induction process.

## Sustainment of SOCE and NFAT nuclear translocation are reduced in tTregs, but not in iTregs

It was reported previously that TCR signaling led to a dramatic reduction of Foxp3+ cells in iTregs, but not in tTregs (*Chen et al., 2011*; *Selvaraj and Geiger, 2007*). We reproduced these data in our laboratory (*Figure 2A*). tTregs have been reported to show an attenuated response to TCR triggering (*Sumpter et al., 2008*; *Vaeth et al., 2012*; *Yan et al., 2015*), the exact point of signal attenuation in tTregs has not been elucidated. In immune cells, there are mainly three types of $Ca^{2+}$ signals, the steady-state low-level Ryr2-based membrane proximal $Ca^{2+}$ 'puff'; the TCR ligation-triggered, inositol triphosphate-mediated IP3R opening on the ER (endoplasmic reticulum) endothelia membrane; and ER $Ca^{2+}$ depletion-induced, sustained $Ca^{2+}$ influx upon SOCE activation (*Guse, 1998*). As the extracellular $Ca^{2+}$ is roughly 4 logs higher than the intracellular, SOCE activation is by far the strongest $Ca^{2+}$ event in T cells (*Figure 2—figure supplement 1A*), and is also distinct from the other $Ca^{2+}$ signaling patterns for its persistent vibration lasting hours to days (*Christo et al., 2015*; *Wülfing et al., 1997*).

We wondered if the reported TCR signal truncation in tTregs was manifested as reduced SOCE activities. We titrated anti CD3/CD28 concentrations, and iTregs lost Foxp3 at the lowest dose required to activate, yet tTregs were resistant to an eightfold increase in both (*Figure 2—figure supplement 1B*). To study the molecular basis for the weak response in tTregs, we first monitored the early SOCE intensity in iTregs and tTregs, we found the first wave of SOCE in iTregs was slightly higher than Tconvs and tTregs, but there was no discernable difference between tTregs and Tconvs (*Figure 2B*). In addition, the phosphorylation of CD3 ζ was similar in Tconv and nTreg immediately after TCR cross-link (*Figure 2—figure supplement 1C*). As full T cell activation requires sustained SOCE activation (*Dolmetsch et al., 1998*), we extended the observation time window. As SOCE signal is synchronized upon initial TCR trigger, each individual T cell SOCE oscillation hours into activation is no longer in unison (*Christo et al., 2015*; *Le Borgne et al., 2016*), we therefore recorded $Ca^{2+}$ waves in individual cells. In Tconvs and iTregs, $Ca^{2+}$ remained cyclic waves expected in SOCE pulsation. In contrast, this $Ca^{2+}$ signal was suddenly lost in tTregs 40–50 min after the activation (*Figure 2C*; *Videos 1–3*).

In contrast to other transcriptional factors, such as NF-κB, activated rapidly in response to short $Ca^{2+}$ elevation, NFATs are known to be triggered only by sustained cytoplasmic $Ca^{2+}$ (*Dolmetsch et al., 1998*; *Dolmetsch et al., 1997*; *Tomida et al., 2003*). Indeed, in contrast to iTregs and Tconvs, tTregs' NFATc1 and c2 failed to translocate into the nucleus (*Figure 2D*). Upon detailed analysis, in iTregs, anti-CD3/CD28 stimulation led a quick drop in cytoplasmic NFAT in 2 hr, with moderate accumulation of NFAT in the nucleus (*Figure 2—figure supplement 1D*). This was followed by continuous synthesis of NFAT in the cytoplasm and accumulation of nuclear NFAT (*Serfling et al., 2006*; *Zhou et al., 2002*).

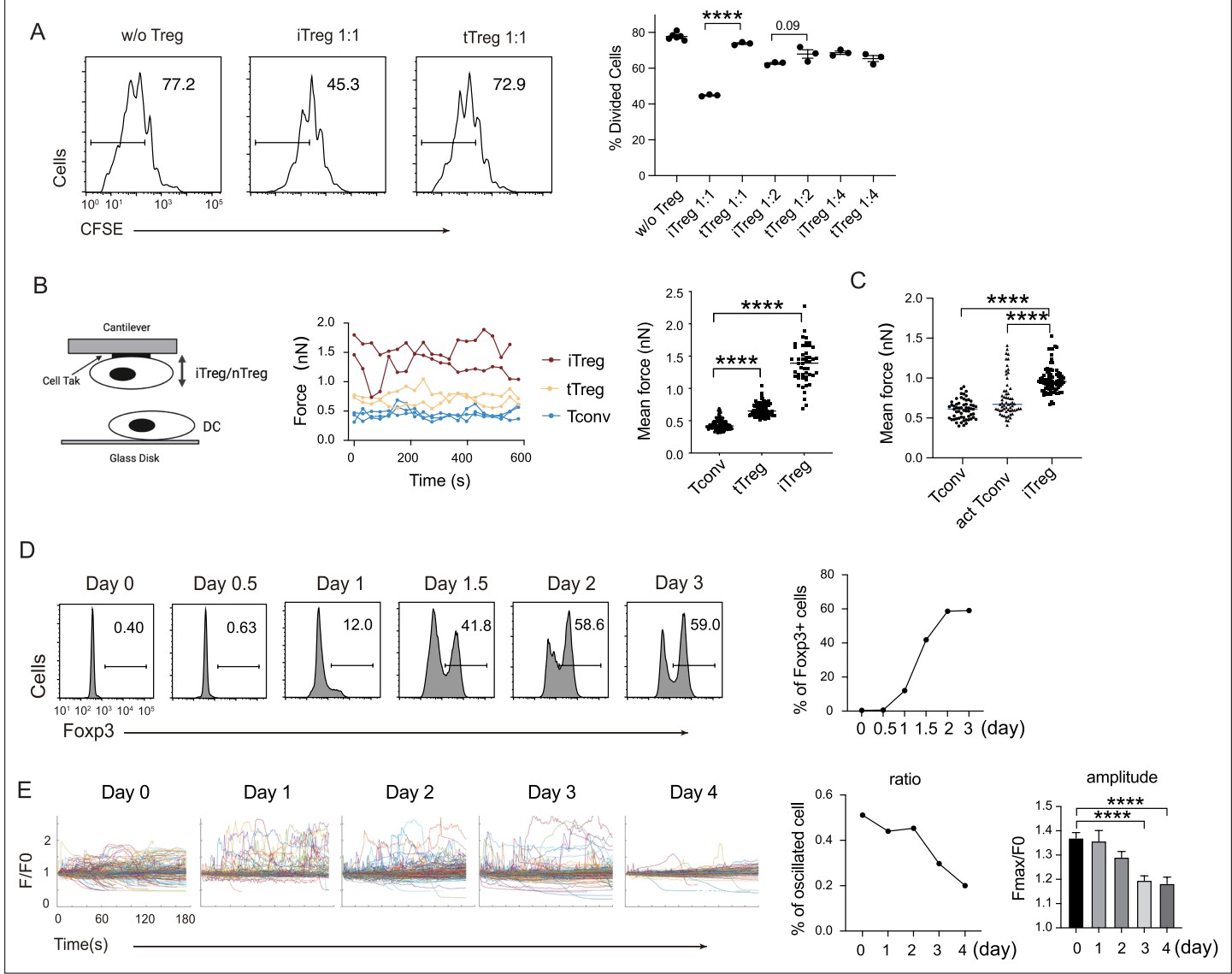

**Figure 1.** iTreg share similar suppressive mechanism to tTregs. (**A**) Comparison of suppressive activity between tTreg and iTreg. CFSE (carboxy fluoroscein succinimidyl Eester) -labeled OT-II T cells were stimulated with OVA-pulsed DC, then Foxp3-GFP⁺ iTregs and tTregs were added to the culture to suppress the OT-II proliferation. After 4 days, CFSE dilution were analyzed. $n = 3$, $N = 3$. Left, representative histograms of CFSE in divided Tconvs. Right, graph for the percentage of divided Tconvs. (**B**) iTregs possessed stronger binding force to DCs than tTreg. A schematic diagram for AFM-SCFS (atomic force microscope-single cell force spectroscopy) assay setup (left). SCFS force readings for Tconv, tTreg, and iTreg adhering to DC2.4 cells, one line represents a pair of T–DC, every dot represents force reading from each contact. Mean force of Tconv, tTreg, and iTregs adhering to DC2.4 cells. (**C**) iTregs showed increased binding force compared with activated Tconvs. Mean force of Tconv, activated Tconv, and iTreg adhering to DC2.4 cells. $n > 45$, $N = 3$. (**D**) Precise expression of Foxp3 was assessed during iTreg induction. Naïve Tconvs were stimulated with anti-CD3 and anti-CD28, in the presence of TGF-β and IL-2 to induce iTregs. Cells were harvested at the indicated time and Foxp3 expression was analyzed by intracellular staining. (**E**) Basal Ca²⁺ oscillation was assessed during iTreg induction. Naïve Tconvs were stimulated with anti-CD3 and anti-CD28, in the presence of TGF-β and IL-2 to induce iTregs. Cells were harvested at the indicated time and loaded with Fluo-4 AM, and Fluo-4 fluorescence over time were recorded with confocal microscope. The change of intracellular free Ca²⁺ concentration over time were shown as $F/F_0$. The ratio of oscillated cells and standard deviation of $F/F_0$ were calculated. $n > 150$, $N = 3$. Here, ****$p < 0.0001$, by Student's $t$-est.

The online version of this article includes the following source data and figure supplement(s) for figure 1:

**Source data 1.** The suppression activity, AFM force of iTreg and nTreg, and calcium oscillation during iTreg induction.

**Figure supplement 1.** Expression of Ryr2 in iTreg and the effect of Ryr2 on iTreg induction.

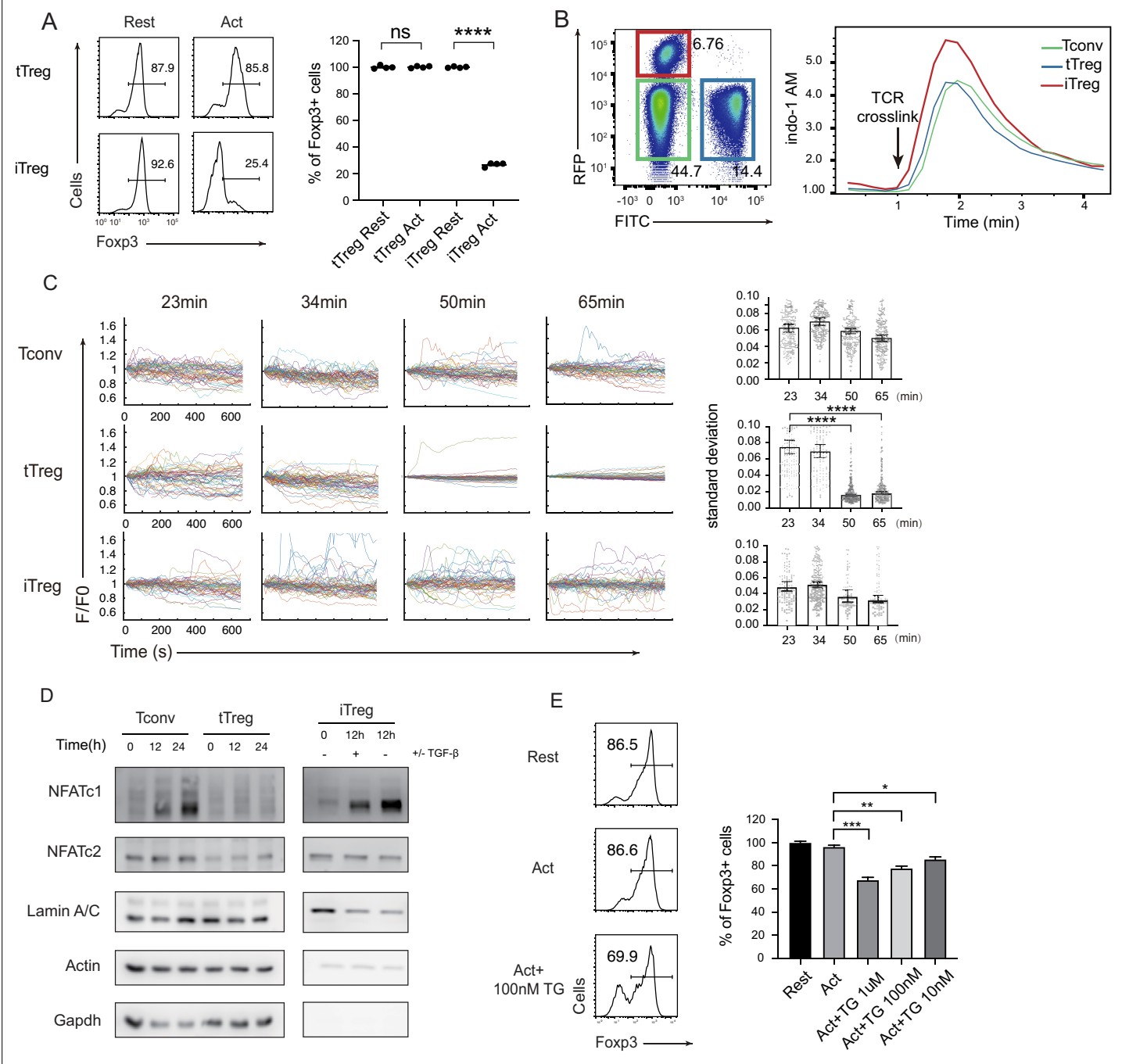

**Figure 2.** Diminished store-operated calcium entry (SOCE) signal and NFAT translocation in tTregs, but not in iTregs. (**A**) Comparison of Treg stability between tTregs and iTregs. iTregs and tTregs were sorted and restimulated with anti-CD3 and anti-CD28 antibodies. Cells were harvested after 2-day restimulation and Foxp3 expression was analyzed by intracellular staining. The percentages of Foxp3+ cells were monitored by FACS (fluorescence-activated cell sorting). Left, representative histograms of restimulated iTregs. Right, graph for the percentage of Foxp3+ cells in all CD4+ cells. $n = 4$, $N = 5$. (**B**) Early SOCE signal was measured in iTreg, tTreg, and Tconv by flow cytometry. Sorted Foxp3-RFP+ iTreg and Foxp3-GFP+ tTreg and double negative Tconv cells were mixed and loaded with Indo-1 AM, then stained with biotin anti-CD3 and biotin anti-CD28 for 1 hr, the baseline fluorescence was recorded for 1 min, and then TCR crosslink was perform by the addition of streptavidin. Left, the gate of three mixed cells. Right, Indo-1 AM ratio of these cells upon TCR crosslink. $N = 3$. (**C**) Long-term SOCE were truncated in tTreg, but sustained in iTreg. Tconv, tTreg, and iTreg cells were loaded with Fluo-4 AM and activated by anti-CD3 and anti-CD28 in confocal dish. Fluorescence was recorded in the indicated time after stimulation with the interval of 10 s. Left, the $F/F_0$ of mean fluorescence intensities were calculated and presented. Right, graph for standard deviation of fluorescence in these cells. $n > 50$, $N = 3$. (**D**) NFAT accumulate much in nucleus of iTreg, but not in tTreg. Tconv, tTreg, and iTreg cells were stimulated by anti-CD3 and anti-CD28, after the indicated times, cells were lysed and the cytoplasmic/nuclear components were separated. The cytoplasmic and nuclear NFATc1

*Figure 2 continued on next page*

DOI: https://doi.org/10.7554/eLife.88874

*Figure 2 continued*

and NFATc2 were analyzed by western blot. Actin and GAPDH were used as loading control of cytoplasmic proteins, and LaminA/C as nuclear. N = 4. (**E**) Forcibly sustained calcium signal destabilizes tTreg. Foxp3-RFP+ tTregs were stimulated by anti-CD3 and anti-CD28. After 1 hr, various concentrations TG were added in the culture medium. Cells were collected after 24-hr stimulation and Foxp3 expression was analyzed by intracellular staining. Left, representative histograms of treated tTregs. Right, graph for the percentage of Foxp3+ cells in all tTregs. n = 3, N = 3. Here, *p < 0.05; **p < 0.01; ***p < 0.001; ****p < 0.0001, by Student's *t*-test.

The online version of this article includes the following source data and figure supplement(s) for figure 2:

**Source data 1.** Treg stability, early store-operated calcium entry (SOCE) signal, long-term SOCE signal in iTreg and nTreg.

**Source data 2.** Original imaging data of long-term store-operated calcium entry (SOCE) signal in Tconv, nTreg, and iTreg, related to *Figure 2C*.

**Source data 3.** Original western data of NFAT nuclear translocation in Tconv, nTreg, and iTreg, related to *Figure 2D*.

**Figure supplement 1.** TCR responsiveness and NFAT translocation in tTregs and iTregs.

**Figure supplement 1—source data 1.** Original western data of NFAT nuclear translocation upon early activation in Tconv, nTreg, and iTreg, related to *Figure 2—figure supplement 1D*.

While tTregs are known to have reduced NFAT activation in response to TCR ligation (*Sumpter et al., 2008*; *Nayak et al., 2009*), our results pointed to a truncated $Ca^{2+}$ response likely to be the molecular basis for this phenotype. With this association potentially established, we wondered if the length of SOCE itself is sufficient to explain the stability difference between iTregs and tTregs. Non-specific $Ca^{2+}$ membrane permeability modifier such as ionomycin resulted in large-scale cell death (*Figure 2—figure supplement 1E*), we therefore used SERCA inhibitor Thapsigargin (TG). Blocking SERCA leads to endosome $Ca^{2+}$ reloading blockage that leads to continuous translocation of STIMs to the plasma membrane–ER junction and Orai channel opening. In contrast to indiscriminating $Ca^{2+}$ overload, pure SOCE activation was not followed by significant cell death (*Figure 2—figure supplement 1E*). As expected, some tTregs treated with TG lost their Foxp3 (*Figure 3E*). These results collectively argue that the truncated $Ca^{2+}$ in tTregs is essential to protect tTregs from phenotypical shift.

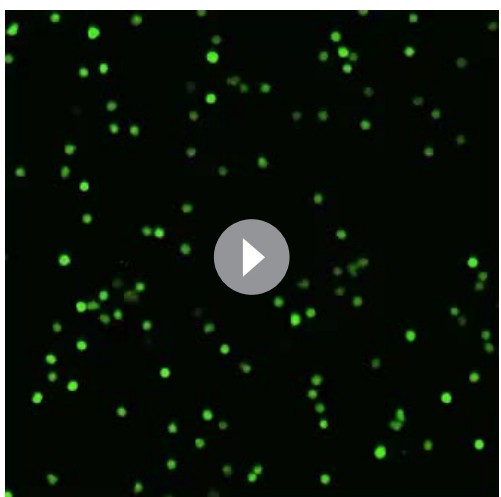

**Video 1.** Imaging of store-operated calcium entry (SOCE) signal in Tconvs. Tconvs (Video 1), tTregs (*Video 2*), and iTregs (*Video 3*) were sorted and resting for 1 day, then cells were loaded with Fluo-4 AM and activated by anti-CD3 and anti-CD28 in confocal dish. After activation for 50min, movies were recorded for total 600s with the interval of 10s.

https://elifesciences.org/articles/88874/figures#video1

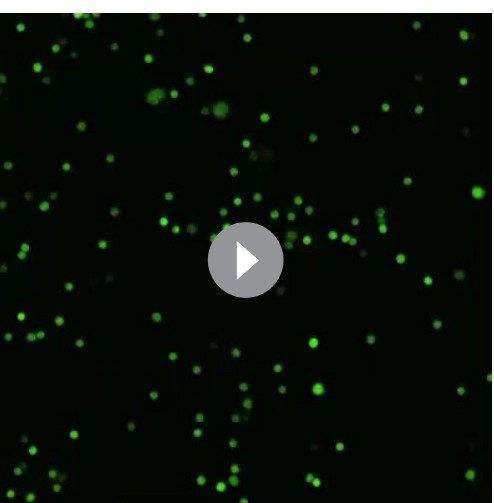

**Video 2.** Imaging of store-operated calcium entry (SOCE) signal in tTregs. Tconvs (*Video 1*), tTregs (Video 2), and iTregs (*Video 3*) were sorted and resting for 1 day, then cells were loaded with Fluo-4 AM and activated by anti-CD3 and anti-CD28 in confocal dish. After activation for 50min, movies were recorded for total 600s with the interval of 10s.

https://elifesciences.org/articles/88874/figures#video2

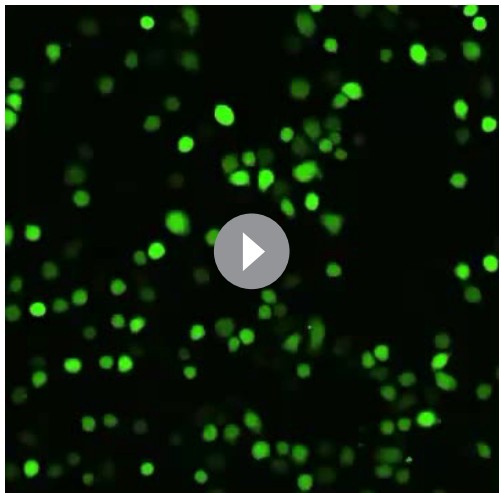

**Video 3.** Imaging of store-operated calcium entry (SOCE) signal in iTregs. Tconvs (**Video 1**), tTregs (**Video 2**), and iTregs (Video 3) were sorted and resting for 1 day, then cells were loaded with Fluo-4 AM and activated by anti-CD3 and anti-CD28 in confocal dish. After activation for 50min, movies were recorded for total 600s with the interval of 10s.

https://elifesciences.org/articles/88874/figures#video3

## iTregs display activated chromatin state with highly open activation region

As shown, the behaviors of SOCE signaling and NFAT of iTregs were closer to their precursor Tconvs rather than tTregs, we were curious about the systemic similarity among iTregs, Tconvs, and tTregs. Therefore, we performed RNA-seq and Assay for Transposase-Accessible Chromatin with high-throughput sequencing (ATAC-seq) of iTregs, together with resting/activated tTregs and Tconvs.

In RNA-seq data, principal component analysis (PCA) and heatmap of DEG (differentially expressed gene analysis) indicated that Tconvs were quite different before and after activation, but the changes of tTregs before and after TCR activation were more limited compared with Tconvs (*Figure 3A* and *Figure 3—figure supplement 1C*), verifying that tTregs have low TCR responsiveness. Comparing the transcriptome characteristics with other cell types, iTregs acquired partial Treg features after induction, shown as an intermediate between Tconvs and tTregs on PC2 (*Figure 3A* and *Figure 3—figure supplement 1B*). While among all cell states, iTregs had the highest PC1 scores, represent higher proliferation state, probably due to the 4 days of induction (*Figure 3A* and *Figure 3—figure supplement 1A*).

Next, we analyzed the chromatin accessibility of these cells using ATAC-seq data. In the PCA plot, tTregs showed fewer changes after activation compared to Tconvs, while iTregs were closer to activated Tconv cells (*Figure 3B*). In differential peak heatmap clustered by cell type, Tconvs became more accessible in 'Activation Region' after activation, and with a dramatic increase in the overall accessibility (*Figure 3C*). However, the changes of tTregs after activation are limited, the majority of tTreg-specific accessible region, named 'Treg Region', were not changed. Also the activation region opening seen in Tconvs were not more accessible in tTregs (*Figure 3C*). All of these characteristics indicated that tTreg phenotype is highly ensured by their low TCR responsiveness which potentially led to their refusal to change chromatin accessibility seen in Tconvs following activation.

When comparing iTregs with the others, we found iTregs were highly accessible in activation region found in activated Tconvs (*Figure 3C, D*). However, only a few Treg-specific peaks shown in Treg region were opened in iTregs (*Figure 3C, E*). For further analysis, we isolated Treg-specific peaks. It was clear that most of them were opened in tTregs but showed little accessibility in iTregs (*Figure 3—figure supplement 1D*). For example, *Foxp3* and *Ctla4*, expressed in both tTregs and iTregs, show high accessibility in resting and activated tTregs, but were closed or only partially opened in iTregs (*Figure 3G*). In contrast, genes related to inflammation cytokine in T helper cells, such as *Ifng*, *Il4*, *Il17ra*, and *Il21*, were highly opened in iTregs (*Figure 3F*). In further elaboration of Activation Region in *Figure 3D*, we have selected some genes including typical peaks that are within this region (*Figure 3—figure supplement 1E*). These genes encompass some T cell activation-associated transcription factors, such as *Irf4*, *Atf3*, as well as multiple members of the Tnf family including *Lta*, *Tnfsf4*, *Tnfsf8*, and *Tnfsf14*. Additionally, genes related to inflammation cytokine and function such as *Il12rb2*, *Il9*, and *Gzmc* are included. These genes display elevated accessibility upon T cell activation, partially open in activated tTreg cells. However, all of them exhibit high accessibility in iTreg cells.

Overall, we showed that at the gene expression level, iTregs have acquired some Treg features, but the chromatin accessibility was mostly trapped in activated Tconv state, not only with regard to the closed Treg genes, but also with genes associated with T activation and inflammation highly opened. The latter is expected to be mainly closed in genuine Tregs in accordance with their lower

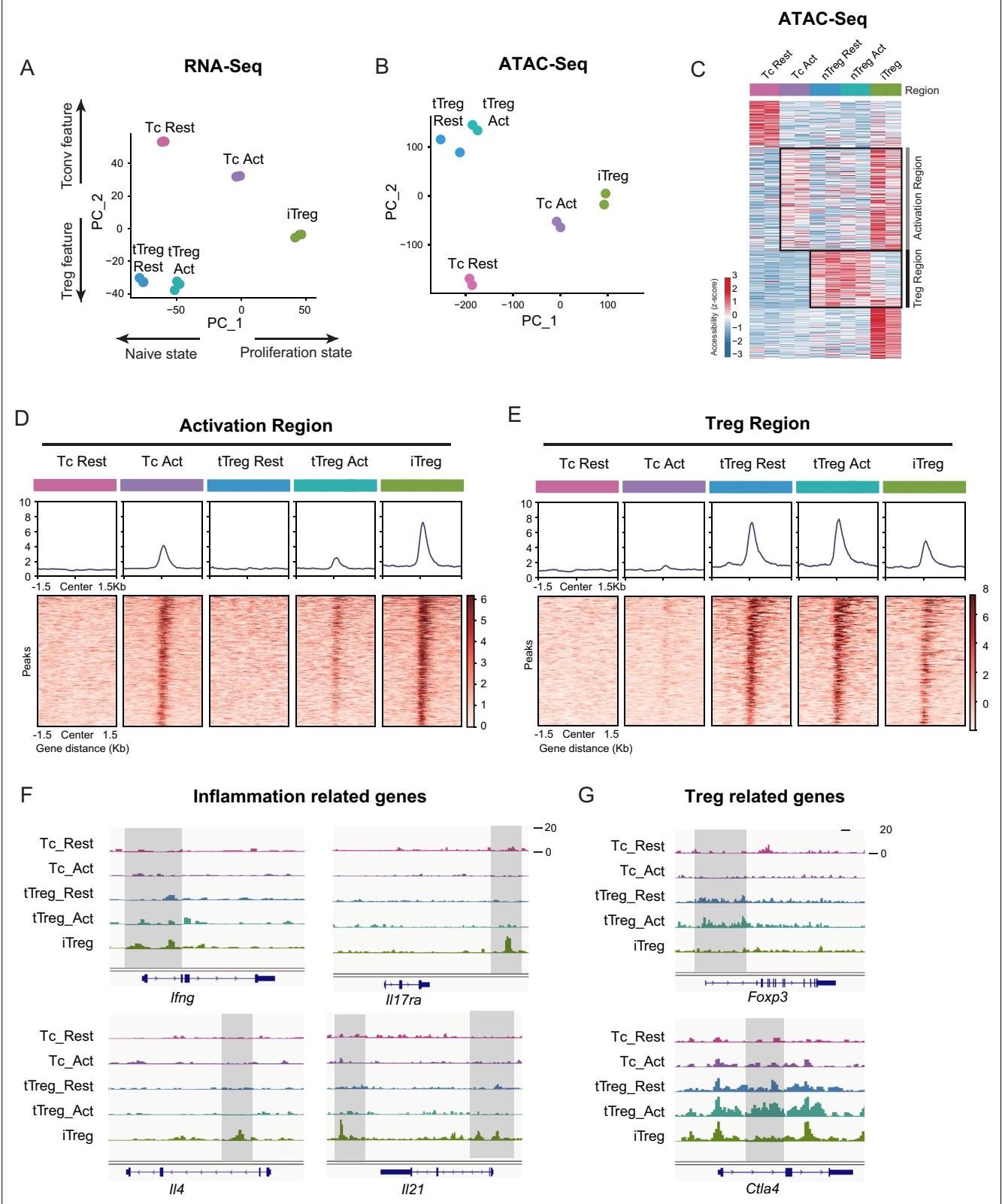

**Figure 3.** iTregs display highly open chromatin state at the activation and differentiation-related genes. (**A**) Principal component analysis (PCA) visualization of transcriptional profiles of Tconvs, tTregs, and iTregs with or without TCR stimulation. Color indicates cell types. (**B**) PCA visualization of chromatin accessibility profiles of different cell types. Color indicates cell type. (**C**) Heatmap showing the chromatin accessibility of cell type specifically accessible peaks. As shown, two major groups of genes were labeled on right. (**D**) Line plots (top) and heatmaps (bottom) of activation regions in

*Figure 3 continued on next page*

*Figure 3 continued*

Tconvs, iTregs, and tTregs. Activation regions were determined by a threshold of adjusted p < 0.05 calculated by DESeq2. (**E**) Line plots (top) and heatmaps (bottom) of Treg regions in Tconvs, iTregs, and tTregs. Treg regions were determined by a threshold of adjusted p < 0.05 calculated by DESeq2. (**F**) Genomic track showing the chromatin accessibility of *Ifng*, *Il4*, *Il17ra*, and *Il21*. (**G**) Genomic track showing the chromatin accessibility of *Foxp3* and *Ctla4*.

The online version of this article includes the following figure supplement(s) for figure 3:

**Figure supplement 1.** iTreg has highest proliferation state and partial Treg feature.

responsiveness to TCR triggering. We speculate that this 'unconverted' chromatin accessibility, especially the highly opened activated genes will contribute to iTreg instability upon TCR stimulation. It is therefore interesting whether NFAT accumulated in nuclei driven by sustained calcium oscillation will indeed target those genes for an activation state unfit for their phenotypical stability.

## SOCE and NFAT directly leads to phenotypical instability in iTregs

To further validate the role of calcium and NFAT in iTreg instability, we used inhibitors cyclosporine A (CsA) and CM-4620 to suppress calcium signal and NFAT, respectively. We found that inhibition of calcium or NFAT rescued the Foxp3 expression in iTregs upon restimulation (*Figure 4A*). Another SOCE inhibitor BTP2 was also verified (*Figure 4—figure supplement 1A*). Accordingly, ionomycin, which can directly induce calcium release independent of TCR activation, also lead to iTreg instability (*Figure 4—figure supplement 1B*). Thus, NFAT and calcium signaling blockage were correlated to iTreg stability. As control, inhibition of NF-κB (BAY 11-7082), c-Jun (SP600125), or a c-Jun/c-Fos complex (T5224) had no discernable effect, or in one case, possibly further reduction in stability (*Figure 4—figure supplement 1C*). These results may indicate that NFAT plays a crucial and special role in TCR activation, which leads to iTreg instability.

To investigate how NFAT and calcium signaling destabilize iTregs, we performed RNA-seq on these cells. From the differential gene expression and PCA, we found that iTregs upon restimulation showed significant differences from the resting state at the transcriptome level, and this difference could be substantially reduced with inhibition of calcium or NFAT (*Figure 4B*). As shown in the PCA plot, calcium/NFAT-blocked iTregs were significantly closer to the resting state than the unstable ones (*Figure 4C*). Thus, inhibition of calcium or NFAT can significantly improve the stability of iTregs as reflected by transcriptomic identity. In addition, the large overlap (about 80%) of genes that were rescued by CM-4620 and by CsA further supported that calcium signaling change the fate of iTregs mainly through NFAT (*Figure 4D*).

Among the genes rescued by CM-4620 and CsA, they can be further divided into two groups. Members of the first group were downregulated upon restimulation and the downregulation was blocked by inhibition of calcium and NFAT. This result indicating that these genes are downregulated by NFAT. Among these genes, several of them are critical for functions and stability of Tregs, such as *Foxp3*, *Id3*, *Il10rb*, *Sema4a*, *Tgfbr1/2*, *Ms4a4b*, and *Nlrp6* (*Figure 4B* and *Figure 4—figure supplement 1D*). In the functional analysis of these genes, we found that Rap-1 pathway, PD-1/PD-L1 pathway and Foxo pathway were significantly enriched (*Figure 4—figure supplement 1E*). Loss of these genes was therefore consistent with the phenotype of iTreg instability.

The other group had genes that were otherwise upregulated after restimulation, but failed to do so with CM-4620 or CsA, indicating that these genes are upregulated by NFAT. In these genes, many of them are associated with T helper differentiation and T cell activation, such as *Il21*, *Il12rb2*, *Tbx21*, *Gzma*, *Stat2*, *Stat3*, *Atf3*, and *Fasl* (*Figure 4B, F*). Among them, *Il21* encodes a key cytokine in Th17 differentiation, *Tbx21*, encoding T-bet, and *Il12rb2* are key transcriptional factors and cytokine receptor for Th1 differentiation. Functional enrichment analysis of these upregulated genes pointed to Th17 related pathway and Th1/Th2 related pathways (*Figure 4E*). Further validation of these upregulation T helper related genes (T$_H$ genes) by NFAT was perform by QPCR (*Figure 4G*). These results in all indicated the iTreg restimulation was associated with a strong tendency of differentiation toward non-Treg polarization, and NFAT was likely a key regulator in this process.

## NFAT upregulate prime-opened T$_H$ genes to destabilize iTregs

To further study the role of NFAT in iTreg instability, we performed Cut&Tag assay to acquire the NFAT-binding sites in resting iTregs and those after restimulation (*Figure 5A*). NFAT-flag and anti-Flag

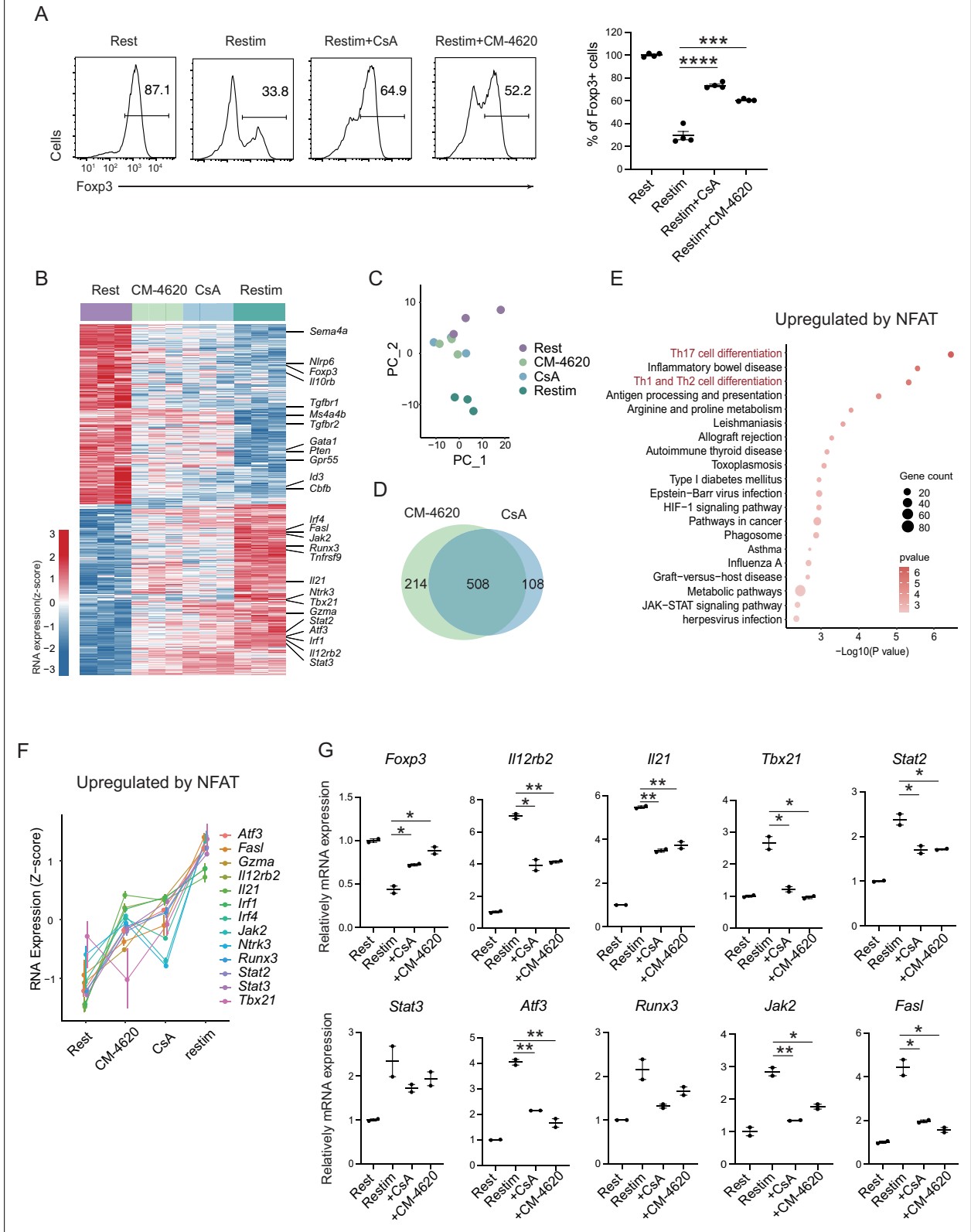

**Figure 4.** Store-operated calcium entry (SOCE) signaling and NFAT can disrupt iTreg stability. (**A**) Impact of calcium signal and NFAT on iTreg stability. Sorted Foxp3-GFP+ iTregs were rested for 1 day, then restimulated by anti-CD3 and CD28 in the presence of cyclosporine A (CsA) and CM-4620. Percentages of Foxp3+ cells were analyzed by intracellular staining after 2-day restimulation. Left, representative histograms of CsA and CM-4620-treated iTregs. Right, graph for the percentages of CsA and CM-4620-treated Foxp3+ cells in all CD4+ cells. $n = 4$, $N = 8$. (**B**) Heatmap showing the

*Figure 4 continued on next page*

*Figure 4 continued*

decreased changes for restimulated iTregs with CM-4620 or CsA. Typical genes were highlighted. (**C**) Principal component analysis (PCA) visualization of transcriptional profiles of iTregs at different states. Color indicates cell states. (**D**) Venn plot showing the overlap of DEGs rescued by adding CM-4620 or CsA. (**E**) Biological terms enriched by the significant upregulated genes after iTreg restimulation rescued by inhibiting calcium or NFAT. (**F**) Representative genes upregulated after iTreg restimulation rescued by inhibiting calcium or NFAT. (**G**) QPCR of *Foxp3* and Th-differentiated gene expression in the resting, restimulated and CsA/CM-4620-treated iTregs. Here, *p < 0.05; **p < 0.01; ***p < 0.001; ****p < 0.0001, by Student's *t*-test.

The online version of this article includes the following source data and figure supplement(s) for figure 4:

**Source data 1.** iTreg stability, QPCR of related gene expression upon activation after CsA and CM-4620 blockade.

**Figure supplement 1.** Store-operated calcium entry (SOCE) signaling and NFAT cause instability and downregulate Treg-related gene.

---

antibody were used to capture peaks that were bound by NFAT (***Figure 5—figure supplement 1A***). The binding peaks were of high quality as indicated by the lower number of peaks in the mock control compared to both resting and activated states (***Figure 5B***). The number of NFAT peaks in restimulated iTregs was significantly higher than that in resting state, which was consistent with the massive nuclear translocation of NFAT indicated by western blot (***Figure 5B*** and ***Figure 2D***). For peak distribution at gene body, NFAT peaks both before and after restimulation were enriched at promoters/transcription start site (TSS) and the first introns of genes (***Figure 5—figure supplement 1C***).

For those upregulated $T_H$ genes, we speculated that this may be associated with the highly 'activated' chromatin state of iTregs mentioned earlier. Therefore, we first checked the chromatin accessibility around the TSS in those genes and found that this group of genes was significantly more accessible in iTregs than in tTregs and Tconvs (***Figure 5C***). As shown in the genome browser view, *Il21*, *Il12rb2*, and *Tbx21* were more open in iTregs, even more so than the activated Tconvs (***Figure 5—figure supplement 1B***). Further analysis revealed that NFAT significantly increased their binding to the TSS regions on upregulated Th genes after restimulation (***Figure 5D***), while showing little change on downregulated Treg genes (***Figure 5—figure supplement 1D***). As shown by *Il21*, *Tbx21*, and *Il12rb2*, after activation, NFAT accumulated at the sites that were prime-opened resulting in their increased expression (***Figure 5E***). Therefore, the enhanced chromatin accessibility revealed in ***Figure 4*** was indeed approached by the large amount of NFAT, leading to T cell polarization toward other directions with a sole reciprocal loss of polarization toward Tregs. This tendency may to some extent explain the rapid loss of iTreg phenotype.

As one potential counter argument, NFAT is ostensibly essential to the development of tTregs (***Li et al., 2014***; ***Zheng et al., 2010***). In the induction of iTregs, NFAT in collaboration with SMAD is also critical (***Vaeth et al., 2012***; ***Zheng et al., 2010***; ***Tone et al., 2008***). It is therefore intriguing why NFAT re-activation after iTreg formation may serve as a key factor in compromising iTreg stability. To resolve those seemingly contradictory regulations, we performed NFAT motif enrichment analysis. The results showed that putative BATF, AP-1(Fos/Jun), and RORg-binding sites were highly enriched in restimulation state specific regions (***Figure 5F***). For the preference of different cofactors to different gene subsets, SMAD family tended to bind to genes that were downregulated by NFAT, for example, Foxp3, while AP-1 and RORg preferred binding to genes that were upregulated by NFAT, such as $T_H$ genes (***Figure 5G***). Taken together, our results appear to suggest that upon restimulation of iTregs, NFAT prefers to cooperate with AP-1/RORg to promote the expression of prime-opened $T_H$ gene expression, and at the same time denies the SMAD programming that may be critical to maintain Treg functions and stabilities. Certainly, how much this selective cooperation between NFAT and other transcriptional factors contributes to iTreg instability remains speculative until large-scale ChIP-seq analyses are performed on the players involved.

According to the Cut&Tag data, the secretion levels of IL-21 in tTreg and iTreg were also examined by ELISA. As shown in ***Figure 5H***, tTreg did not secrete IL-21 regardless of activation status (undetectable), while iTreg did not secrete IL-21 at resting state but they did so after 48 hr of restimulation (***Figure 5H***). The secretion of IL-21 was inhibited by CsA and CM-4620 treatment. This observation aligns with our findings where nuclear binding of NFAT to gene loci of these cytokines enhanced their expression, pushing iTreg unstable under inflammatory conditions. These findings further underscore the likelihood that the inhibition of calcium and NFAT signaling might contribute to the stabilization of iTreg by suppressing the secretion of inflammatory cytokines.

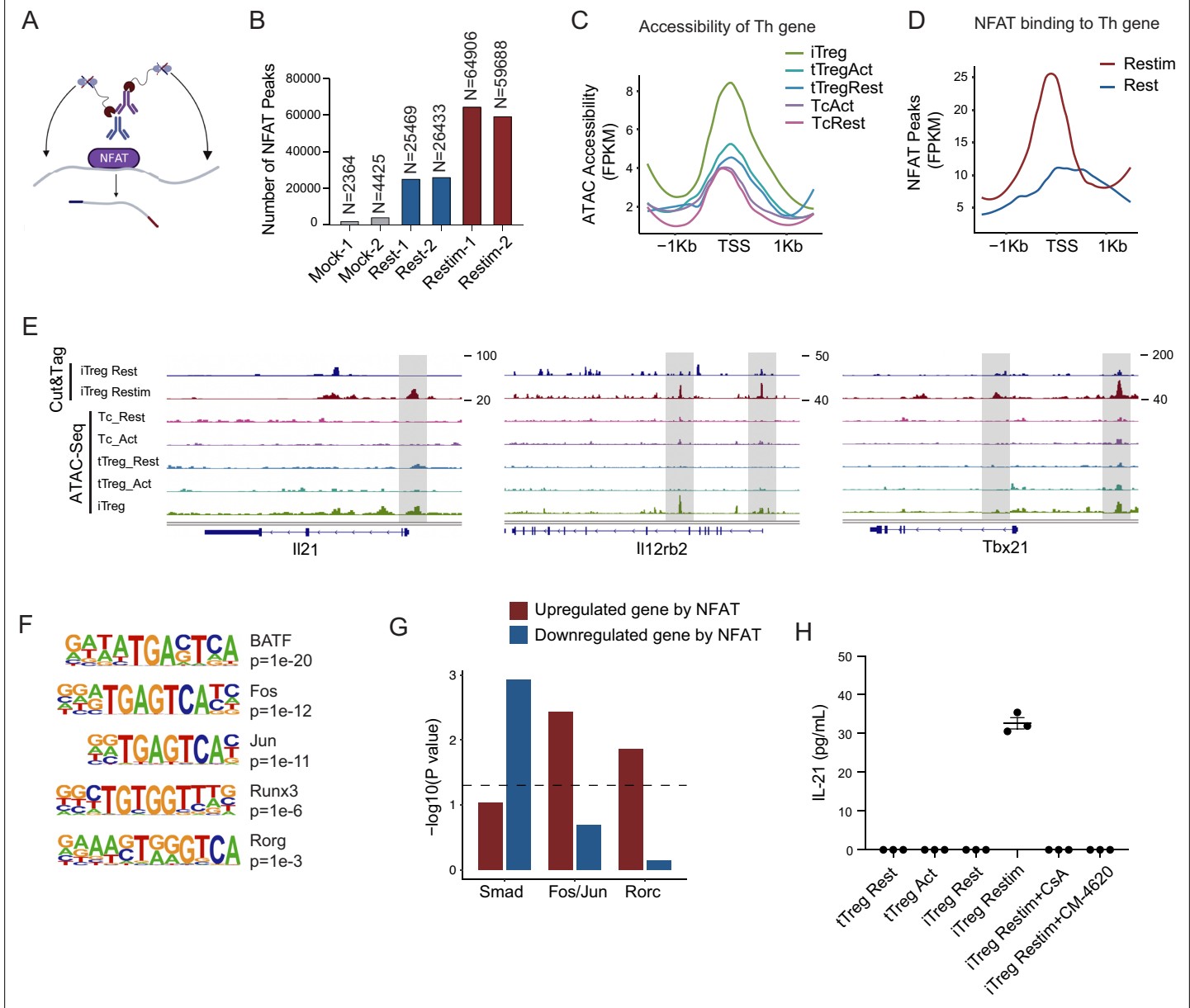

**Figure 5.** NFAT disrupt iTreg stability by upregulating prime-opened $T_H$ genes. (**A**) Model of CUT&Tag experiments to capture the binding sites of NFATc1. (**B**) The number of NFAT Cut&Tag peaks in mock control, resting iTregs, and restimulated iTregs. (**C**) Normalized counts of ATAC-seq reads in resting and activated Tconvs, resting and activated tTregs and iTregs, centered on the transcription start site (TSS) region of NFAT-upregulated $T_H$ genes. (**D**) Normalized counts of NFAT Cut&Tag reads in resting and restimulated iTreg, centered on the TSS region NFAT-upregulated $T_H$ genes. (**E**) Genome track visualization of NFAT-binding profiles and chromatin accessibility profiles in typical genes *Il21*, *Il12rb2*, and *Tbx21*. (**F**) Motif enriched in peaks with higher NFAT Cut&Tag signals in restimulated iTreg versus resting iTreg. List of five representative motifs ranked based on the p-values. The enrichment was performed by using HOMER. (**G**) The enrichment of motif occurrence for typical NFAT cofactors Smad, Fos/Jun, Rorc in the NFAT peaks in NFAT-upregulated or -downregulated genes after restimulation. p-value was from Fisher's exact test. (**H**) IL-21 secretion in tTreg and iTreg upon activation. iTregs and tTregs were sorted and restimulated with anti-CD3 and anti-CD28 antibodies, in the presence of cyclosporine A (CsA) and CM-4620. Cell culture supernatants were harvested after 2-day restimulation and IL-21 secretion was analyzed by ELISA. *n* = 3.

The online version of this article includes the following source data and figure supplement(s) for figure 5:

**Source data 1.** IL-21 secretion in tTreg and iTreg upon activation.

**Figure supplement 1.** iTregs have highest accessibility in $T_H$-associated genes.

## Reduction of SOCE enhances iTreg stability and suppressive potential

Given that sustained calcium signal and NFAT translocation can directly disrupt iTreg stability, we asked whether manipulation of calcium signaling or NFAT can conversely be used to stabilize those cells. As Treg suppression assays require the participation of other T cells which are sensitive to inhibitory agents added indiscriminately, after regular iTreg induction, we overexpressed a dominant negative ORAI (DN-ORAI), whose single monomer incorporation will lead to a functionally disabled ORAI hexamer, to control the SOCE signal upon iTreg restimulation (*Mignen et al., 2008*). As expected, the mutated ORAI reduced the SOCE signaling by about a half (*Figure 6A*). When we restimulated this DN-ORAI expressing iTregs with anti-CD3/CD28, they showed stronger stability with higher retention of Foxp3 than WT-ORAI iTregs (*Figure 6B*). Consistently, DN-ORAI iTregs displayed more potent suppressive activity than control iTreg in in vitro suppression assay (*Figure 6C*).

Finally, the stability of DN-ORAI iTregs in vivo was examined. In brief, OT-II iTregs were infected with DN-ORAI retrovirus and then purified by GFP fluorescence, then the CD45.2+ GFP+ Foxp3-RFP+ iTregs were adoptively transferred into CD45.1 recipient, and OVA/Alum were administrated to restimulate DN-ORAI iTregs in vivo. Analysis of the percentage of Foxp3+ cells in mLN 5 days after adoptive transfer showed that DN-ORAI iTregs were more resistant to the loss of Foxp3 and thus were more stable than control WT-ORAI iTregs (*Figure 6D*). These results collectively showed that weakened calcium signal and inhibition of NFAT can quantitatively prevent iTregs from losing more Foxp3 both in vitro and in vivo, thus permitting a stronger suppressive capacity and a longer stability.

Based on extensive opening of activation regions approached by NFAT in the process iTreg stimulation, we postulate that openness of activation region could be another crucial feature of Treg stability, in addition to closed Treg region by low TSDR (Treg-specific demethylated region). Beyond the basal regimens of anti-CD3, anti-CD28, IL-2, and TGF-β, several additional factors have been reported to enhance the stability of iTregs, including the use of retinoic acid, rapamycin, vitamin C, and AS2863619 as well as removal of CD28 signaling (*Akamatsu et al., 2019*; *Hill et al., 2008*; *Mikami et al., 2020*; *Yue et al., 2016*; *Zhang et al., 2013*). We observed if those new additions in some ways altered the state of TSDR as well as the the openness of activation region. Our findings revealed that rapamycin and AS tended to close the activation region (*Figure 6—figure supplement 1A*), while Vc, removal of CD28, and AS increased the accessibility of Treg genes (*Figure 6—figure supplement 1B*). These results indicate that different treatments have varied impacts on the activation region and Treg region. Hence, the selective and combinatorial use of those factors is an interesting proposal in the generation of more stable iTregs.

## Discussion

tTregs are different from other subsets of CD4 T cells in the thymic determination of their phenotype. To some extent, this centric determination is closer to CD4/CD8 demarcation, rather than other polarizations in the peripheral environment (*Rudensky, 2011*; *Sakaguchi et al., 2008*). The stable phenotype, even after activation, is critical to provide the baseline immune suppression (*Rubtsov et al., 2010*). pTregs/iTregs are driven by the environment. It is logical that those cells should be deprived of their suppressive ability when are no longer needed to avoid unnecessary accumulation of those cells. The current understanding is that during thymic development, *Foxp3* CNS2 region is selectively demethylated, allowing several transcriptional factors, such as Ets1 and Foxp3 itself to binding to this location (*Samstein et al., 2012*; *Ohkura et al., 2012*). This is one of several epigenetic features that maintain the stability, and TCR signaling in tTregs does not change the pre-established hypomethylation state. Induced Tregs, on the other hand, use a different program that relies on the binding of SMADs to the CNS1 region to initiate *Foxp3* expression (*Tone et al., 2008*). This SMAD binding mediated transcription is dependent on *Foxp3* H3K4me3 modification. In the absence of SMAD, H3K4me3 is partially lost, resulting in iTreg phenotypical shift (*Tone et al., 2008*). Withstanding this paradigm, iTregs do not immediately lose Foxp3 expression upon TGF-β withdrawal, the loss of Foxp3 is evident hours or days after TCR signaling (*Selvaraj and Geiger, 2007*). This indicates that while the epigenetic alterations are necessary, an activation event is required to drive the loss of phenotype. This notion is subtly echoed by the limited TCR signaling in tTregs, implying that the hyporesponsiveness to TCR ligation may be a critical feature to maintain their stability.

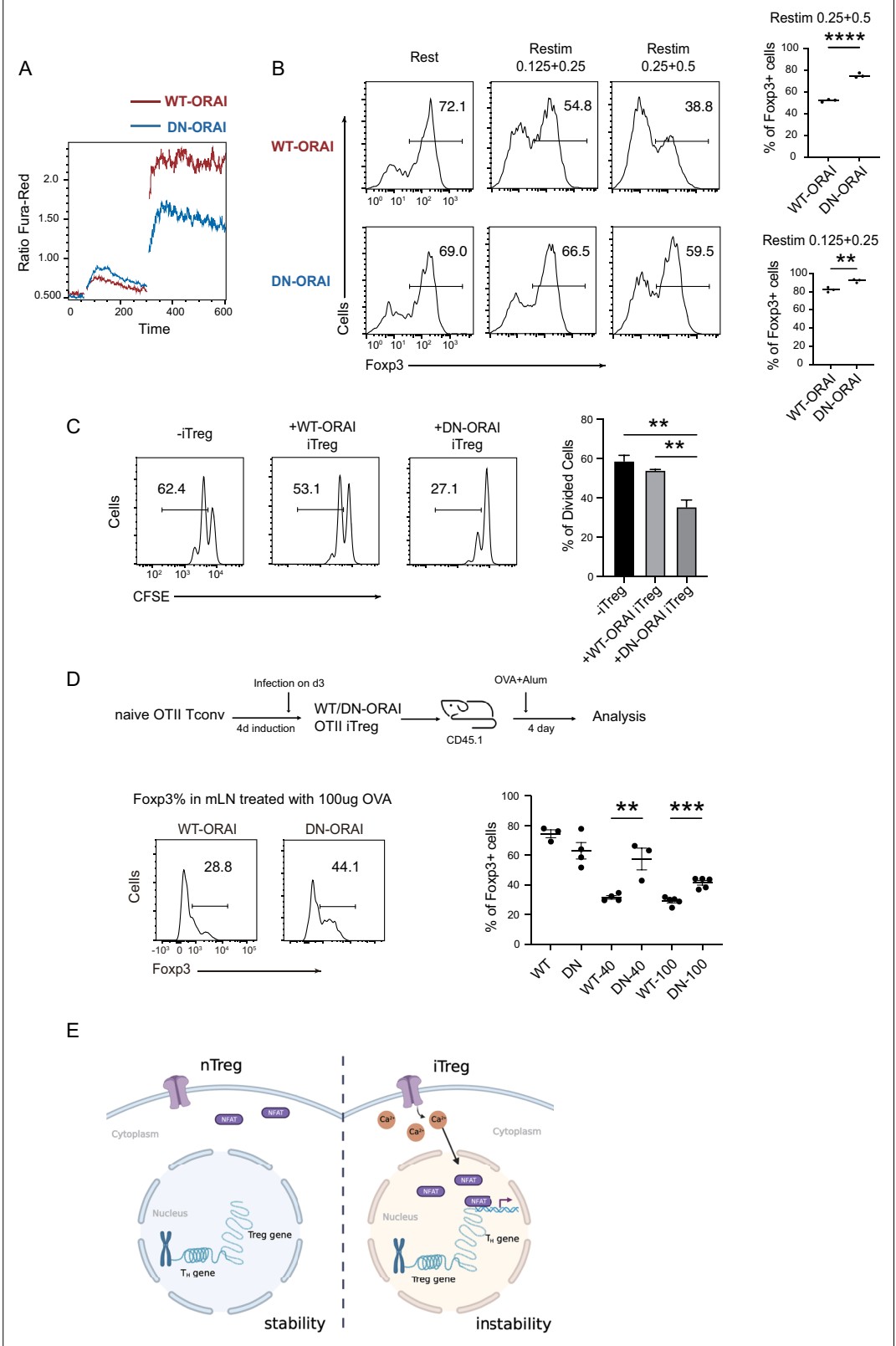

**Figure 6.** Manipulation of store-operated calcium entry (SOCE) can enhance iTreg stability. (**A**) Manipulation of SOCE signal by dominant negative ORAI (DN-ORAI). SOCE was recorded in DN-ORAI iTreg cells loaded with Fura-Red by flow cytometry, TG was added after 1 min to induce ER depletion, 5 min later 2 mM calcium was added to induce calcium influx. $N$ = 3. (**B**) Stability of iTreg was enhanced by DN-ORAI. iTregs were infected with

*Figure 6 continued on next page*

*Figure 6 continued*

WT-ORAI and DN-ORAI, and then restimulated by anti-CD3 and CD28 for 2 days. Percentages of Foxp3 were analyzed by intracellular staining. Left, representative histograms of Foxp3 expression in DN-ORAI iTregs. Right, graph for the percentages of Foxp3$^+$ cells in all CD4+ cells. $n$ = 3, $N$ = 3. (**C**) DN-ORAI enhances iTreg suppressive capacity. CFSE-labeled OT-II T cells were stimulated with OVA-pulsed DC, Foxp3-GFP$^+$ WT-ORAI/DN-ORAI iTregs were added to the culture to suppress the OT-II proliferation. After 40 hr, CFSE dilutions were analyzed. $n$ = 3, $N$ = 3. (**D**) Stability of DN-ORAI iTreg in vivo. WT-ORAI/DN-ORAI-GFP$^+$-transfected CD45.2$^+$ Foxp3-RFP$^+$ OT-II iTregs were transferred i.v. into CD45.1 mice. Recipients were immunized with OVA323-339 in Alum adjuvant. On day 5, mLN and spleen were harvested and analyzed for Foxp3 expression by intracellular staining. Up, schematic representation of adoptive transfer experiment. Bottom right, histograms of Foxp3 expression in CD4$^+$CD45.1$^+$ cells in spleen and mLN; bottom left, graph for the percentages of Foxp3$^+$ cells in all CD45.2$^+$ cells in mLN. $n$ = 3, $N$ = 3. (**E**) Proposed model. The diminished calcium signal and closed chromatin structure in tTregs protect them from genetic and epigenetic disturbances, and the sustained calcium signal in iTregs cause NFAT aggregation in the nucleus, which makes use of a pre-opened gene loci to upregulate T$_H$ genes, thus resulting the instability of iTregs. Here, **p < 0.01; ***p < 0.001; ****p < 0.0001, by Student's *t*-test.

The online version of this article includes the following source data and figure supplement(s) for figure 6:

**Source data 1.** Store-operated calcium entry (SOCE) signal, iTreg stability in vitro and in vivo after over-expression DN-ORAI.

**Figure supplement 1.** Different optimization of iTreg have various impact on activation and Treg regions.

We observed differences between tTregs and iTregs in calcium signal and Th gene accessibility. tTregs utilize both truncated calcium signal and conserved chromatin accessibility to systematically reduce their TCR responsiveness, ensuring their stability. In contrast, iTregs retain the full strength of the TCR signaling, particularly the sustained vibration of SOCE signal, the key factor that drives NFAT synthesis and nuclear translocation. With the help of several cofactors, such as AP-1, NFAT caused the high expression of these T$_H$ genes, which subsequently subverts the original differentiation of iTregs, leading to their instability. Direct inhibition of SOCE signaling and NFAT can enhance iTreg stability in vitro and in vivo (*Figure 6E*).

Previous observations also showed that the calcium signal in tTreg is weaker than that in Tconvs, and NFAT activity is suppressed in tTregs. The weaker Ca$^{2+}$ signal in previous reports was seen as early as 10 min into activation, this contradiction likely reflects the experimental settings and stimulation strength (*Yan et al., 2015*). Nevertheless, in line with our results, inhibition of NFAT by low-dose CsA increased the number of tTreg in atopic dermatitis patients (*Brandt et al., 2009*). It seems reasonable to establish an inverse association between SOCE signal and Treg stability. In our analysis of the RNA-seq and chromatin accessibility data, we did not identify any particular mechanism that may account for the distinct Ca$^{2+}$ signal patterns in tTregs and iTregs. However, in contrast to Tconvs and iTregs, there must be a negative signal loop that blunts SOCE vibration in tTregs. We tried to sustain the calcium signaling in tTregs forcibly by thapsigargin, and indeed observed loss of Foxp3. However, as we at the moment cannot appreciate the finesse of this regulation, the underlying molecular details will require much more delicate approaches rather the brutal force of ER store depletion. In our view, future investigation in this direction may lead to better understanding of Treg biology as well as innovations in clinical Treg cell therapy.

Our results are consistent with the understanding that Treg-specific hypomethylation is crucial for tTregs to stably express their Treg-specific genes (*Samstein et al., 2012*; *Ohkura et al., 2012*). Those epigenetic changes are mostly absent in iTregs, resulting in their instability. The differences of accessibility in Treg genes were also observed in our ATAC-seq data. Therefore, current treatments such as vitamin C and CD28 blockage applied to achieve demethylation and accessibility of Treg genes in iTregs are worthy attempts (*Mikami et al., 2020*; *Yue et al., 2016*). However, our work indicates that at least in ATAC-seq analyses, there are a cluster of activation and differentiation-related peaks present in iTregs that remain silent in tTregs upon TCR stimulation. Critically, these peaks were targeted by abundant NFAT in the iTreg nuclei following activation, leading to the upregulation of Th genes. This tendency of deviation away from Tregs may be another key player to iTreg instability. Our effort to produce highly stable iTregs would likely require a combined approach to demethylate Treg-specific genes and at the same time to block the activation and differentiation-related genes. In our

work, targeting NFAT signaling during the restimulation phase appeared to be a potential method. Other ways that can directly close these genes will also likely be beneficial.

In this work, we stress the importance of NFAT signal in the instability of iTregs. Previous research has also unveiled the adverse effects of other TCR signaling pathways on Treg stability, such as NF-κB and mTOR. As all those factors may contribute to the shift in phenotype, it is out of scope of this report to directly compare all those factors. However, the value of this report is the identification of a key difference in $Ca^{2+}$ signaling under TCR ligation that may explain the stability differences between iTregs and tTregs, which may open the door for additional regulations given the key role of SOCE activity in T cell activation.

## Methods

### Mice

All mice were in CD45.2 C57BL/6 background unless noted otherwise. OT-II-transgenic mice and CD45.1 mice were purchased from Jackson Laboratories. Foxp3-IRES-GFP transgenic mice (B6. Cg-Foxp3$^{tm2Tch}$/J, Strain #:006772 from Jackson Laboratory) were a gift from Dr. Hai Qi, School of Medicine, Tsinghua University. Foxp3-RFP mice were a gift from Dr. Xuyu Zhou, Institute of Microbiology, Chinese Academy of Sciences (CAS). Wild-type mice were purchased from Beijing Vital River Laboratory Animal Technology Co., Ltd. All mice were bred and housed at Tsinghua University Animal Facilities and maintained under specific pathogen-free conditions. This study was performed in strict accordance with the recommendations in the Guide for the Care and Use of Laboratory Animals of the National Institutes of Health. All of the animals were handled according to approved Institutional Animal Care and Use Committee (IACUC) protocols of the Tsinghua University. The protocol was approved by the Committee on the Ethics of Animal Experiments of the Tsinghua University (22-SY6). All surgery was performed under sodium pentobarbital anesthesia, and every effort was made to minimize suffering.

### Antibodies and reagents

Recombinant human IL-2 and human TGF-β were from R&D systems. Anti-mouse CD3 and biotin-anti-CD3 (145-2C11) mAb, anti-mouse CD28, and biotin-anti-CD28 (37.51) mAb purified streptavidin were all from biolegend. LIVE/DEAD Fixable Aqua were from Invitrogen. For flow cytometric analysis, in addition to CD45.1 (A20) monoclonal antibody from Invitrogen, others were purchased from eBioscience or BD Pharmingen. The following clones were used: Foxp3 (150D/E4), CD45.2 (104), CD4 (GK1.5), CD25 (PC61.5), CD44 (IM7), and CD62L (MEL-14). Antibody used in immunoblot including NFATc1 Antibody (7A6) and NFATc2 Antibody (4G6-G5) were from Santa Cruz, Lamin A/C (4C11), β-Actin (13E5), and GAPDH (D4C6R) mAb were from CST. Monoclonal ANTI-FLAG M2 antibody used in Cut&Tag assay was from Sigma. Mouse CD4 T Cell Isolation Kit used for the enrichment of CD4+ T cells was from Stem cell. Foxp3/Transcription Factor Staining Buffer Set for Foxp3 staining was from eBioscience. NE-PER Nuclear and Cytoplasmic Extraction Reagents used for immunoblot was from Thermo. All sequencing kit, including TruePrep DNA Library Prep Kit V2, Hyperactive Universal CUT&Tag Assay Kit, and TruePrep Index Kit V2, was from Vazyme. For calcium mobilization, Ionomycin was from Beyotime and thapsigargin from Invitrogen. For inhibitors, cyclosporin A, CM-4620, and BTP2 were all from MCE. OVA$_{323-339}$ peptides from MBL. For calcium imaging, Fluo-4 AM was from Thermo and Indo-1 AM from BD.

### Flow cytometry sorting and FACS

For surface marker detection, cells were incubated with Fc blocker (CD16/32 antibody; 2.4G2) for 5 min, and then incubated with surface antibody for 15 min at room temperature avoiding light. Foxp3 intracellular staining was performed by cell fixation and permeabilization using the Foxp3/Transcription Factor Staining Buffer Set. Samples were analyzed on Fortessa cytometers (BD) with FACS Diva software and data were analyzed with Flowjo.

To prepare Tconv, tTreg, and iTreg for experiments, spleen was isolated from 2- to 4-month-old Foxp3-GFP mice for Tconv and tTreg sorting, and 6-week-old mice for iTreg induction. Subsequently, CD4+ T cells were enriched using CD4 Isolation kit and stained with indicated surface markers. FACSAriaII (BD) was used for collecting particular population. The cell populations used are as follows:

naive T cells, CD4$^+$ Foxp3-GFP$^+$ CD44$^{low}$ CD62L$^{high}$; tTreg cells, CD4$^+$ Foxp3-GFP$^+$; iTreg, Foxp3-RFP$^+$ or Foxp3-GFP$^+$. Infected iTreg, Foxp3-RFP$^+$ MSCV-GFP$^+$.

## In vitro iTreg induction and restimulation

CD4$^+$Foxp3-GFP$^-$ CD62L$^{hi}$CD44$^{lo}$ (or Foxp3-RFP$^-$) naive T cells were sorted using 6-week-old mice and stimulated with plate-bound 0.5 µg/ml anti-CD3 and 1 µg/ml anti-CD28 in the presence of 2 ng/ml human TGF-β and 200 U/ml rhIL-2. iTreg was induced for 4 days and sorted using Foxp3-GFP/RFP fluorescence, then cells were rested in fresh RPMI complete medium containing 200 U/ml rhIL2 for 1 day.

For iTreg restimulation, iTreg was harvested plated at $0.4 \times 10^6$ cells for restimulation with plate-bound 0.5 µg/ml anti-CD3 and 1 µg/ml anti-CD28 in 96 plate. After 2-day restimulation, Foxp3 was analyzed by intracellular staining.

## Calcium measurement

For calcium influx within 10 min upon TCR activation, sorted Foxp3-RFP$^-$ Tconv, Foxp3-RFP$^+$ tTreg and Foxp3-GFP$^+$ iTreg were resting for 1 day, then mixed and loaded with 1 µM Indo-1 AM at 37°C for 30 min in RPMI medium without fetal bovine serum (FBS). Cells were then washed for twice in cold RPMI medium containing 1% FBS, followed by staining with biotinylated anti-CD3 and biotinylated anti-CD28 on ice for 1 hr. For calcium measurement, cells were heated in 37°C water bath for 5 min, then the baseline was recorded for 60 s on LSR II(BD). Calcium influx was triggered by the addition of 100 µg/ml streptavidin to crosslink TCR. Data were analyzed using kenetics in FlowJo.

For calcium oscillation within 1 hr, Foxp3-RFP$^-$ Tconv, Foxp3-RFP$^+$ tTreg, and Foxp3-RFP$^+$ iTreg were sorted and rested for 1 day, then cells were loaded with Fluo-4 AM at 37°C for 30 min. After washing twice with phosphate-buffered saline and further incubating in RPMI medium for 10 min, cells were activated by plate-bound 0.5 µg/ml anti-CD3 and 1 µg/ml anti-CD28 in poly-L-lysine-coated confocal dish. Excess non-adherent cells were removed by flushing with buffer solution after 10 min. Fluorescence were recorded in the indicated time after stimulation as a time lapse for 20 min with an interval of 6 s. The mean fluorescence intensity changes over time in single cells were analyzed using Surface in Imaris and normalized to resting fluorescence $F_0$ (Fluo-4 $F/F_0$), further the standard deviation of $F/F_0$ in single cells was calculated and presented using GraphPad.

## Immunoblot

iTreg, tTreg, and Tconv were rested for 1 day in RPMI medium containing rhIL-2 and treated with CsA for 30 min before activated. Then these cells were stimulated by plate-bound 0.5 µg/ml anti-CD3 and 1 µg/ml anti-CD28 for indicated time, in the absence or presence of TGF-β. After stimulation, cells were immediately lysed using CERI, then nuclear and cytoplasmic component were separated using NE-PER Nuclear and Cytoplasmic Extraction Reagents. After being mixed with sodium dodecyl sulfate loading buffer and boiled for 10 min, the cytoplasmic and nuclear proteins equal to $5 \times 10^5$ iTreg and $2 \times 10^6$ tTreg/Tconv were loaded onto 7.5% PAGE Gels. Then the protein was transferred onto a NC membrane and immunoblotted with NFATc1 and NFATc2, Actin and GAPDH were used as loading control of cytoplasmic protein, and LaminA/C as nuclear. Finally, the immunostained bands were detected by Super ECL Detection Reagent.

## Treg cell suppression assay in vitro

$10^4$ purified DCs from splenocytes were pulsed with 2 µg/ml OVA$_{323-339}$ peptide, then iTreg and tTreg ($2 \times 10^4$) were added onto DCs to occupy the latter for 30 min. CD25$^-$ OTII Tconv cells were sorted and stained by CFSE (Thermo Fisher Scientifc). $2 \times 10^4$ OTII Tconv cells were mixed in DC-Treg culture in a 96-well U bottom plate. The proliferation of OTII T cells was assessed after 2–4 days by CFSE dilution on flow cytometry.

## DN-ORAI and NFAT-flag construction and retrovirus infection

For DN-ORAI, ORAI CDS sequences were amplified from cDNA library and mutated with primer. Then WT-ORAI and DN-ORAI were linked with MSCV-GFP vector. For NFAT-Flag, Nfatc1-flag CDS was amplified from TetO-FUW-flag-Nfatc1 plasmid, and further linked with MSCV-GFP vector. For retrovirus production, platE cells were cultured with Dulbecco's modified Eagle medium containing

10% FBS to 80–90% confluence. For each 10 cm dish, 10 µg MSCV plasmid and 10 µl Neofect transfection reagent were mixed in 500 µl Opti-MEM medium. After 30-min incubation, the DNA mixture were added into the well and gently mixed. After 48 and 72 hr, supernatant containing viruses was collected, filtered with 0.45 µm membrane. 8 µg/ml polybrene was added to virus supernatant, then Foxp3-RFP+iTreg was infected on the days 2 and 3 of induction by centrifugation at $1500 \times g$, 37°C 2 hr. On the day 5 of induction, the infected cells with GFP fluorescence and Foxp3-RFP iTreg was sorted and used.

## In vivo iTreg stability

CD45.2[+] Foxp3-RFP[+] iTreg was sorted, and washed twice. $5 \times 10^6$ cells were adoptively transferred into CD45.1 mice via tail vein. For DN-ORAI iTreg, CD45.2[+] Foxp3-RFP[+] iTreg was infected and GFP[+]RFP[+] iTreg was sorted to transfer to CD45.1 recipient. Six hours after cell transfer, mice were immunized with OVA protein mixed with Alum adjuvant in each hind flank. After 4 days of immunization, mLN were collected and the percentage of Foxp3+ cells in CD45.2[+] was analyzed by intracellular staining.

## AFM-based single-cell force spectroscopy

The experiments were performed as previously described using a JPK CellHesion unit (*Chen et al., 2017*; *Flach et al., 2011*). In brief, to measure Treg-DC adhesion forces in bicellular system, DC2.4 cells were cultured on untreated glass disks. iTreg/tTreg/Tconv cells were sorted and rested with rhIL2 overnight. The disks were moved into an AFM-compatible chamber and mounted on to the machine stage. The incubator chamber in which the machine was housed was conditioned at 37°C and at 5% $CO_2$. A clean cantilever was coated with CellTak (BD), and then used to glue individual iTreg/tTreg/Tconv cells added to the disk. Only round and robust cells were selected for AFM gluing. The AFM cantilever carrying a single iTreg/tTreg/Tconv cell was lowered to allow T cell contact with an individual DC and to interact for 15 s before being moved upwards, until two cells were separated completely. The force curves were acquired. The process was then repeated. For each SCFS experiment, a pair of T–DC was used to generate force readings from each up and down cycle over a period of several minutes, a minimum of 14 force curves were collected in each T–DC pair. In all experiments, at least three such pairs were used for each condition. The force curves were further processed using the JPK image processing software.

## RNA-seq

The RNA-seq library preparation was performed by a modified SMART-seq2 protocol (*Habib et al., 2016*; *Picelli et al., 2014*). Briefly, RNA was added into 4 µl elution mix made of 1 µl RT primer (10 µM), 1 µl dNTP mix (10 mM each), 1 µl RNase inhibitor (4 U/µl), and 1 µl $H_2O$. Eluted samples were incubated at 72°C for 3 min and immediately placed on ice. Each sample was added with 7 µl reverse transcription (RT) mix made of 0.75 µl $H_2O$, 0.1 µl Maxima RNase-minus RT (Thermo Fisher Scientific, #EP0741), 2 µl 5× Maxima RT buffer, 2 µl Betaine (5 M, Sigma-Aldrich, B0300), 0.9 µl $MgCl_2$ (100 mM), 1 µl TSO primer (10 µM), and 0.25 µl RNase inhibitor (40 U/µl). The RT reaction was incubated at 42°C for 90 min and followed by 10 cycles of (50°C for 2 min, 42°C for 2 min), then heat inactivated at 70°C for 15 min. Samples were then amplified with an addition of 14 µl PCR mix made of 1 µl $H_2O$, 0.5 µl ISPCR primer (10 µM), 12.5 µl KAPA HiFi HotStart ReadyMix (KAPA Biosystems, KK2602). The PCR reaction was performed as follows: 98°C for 3 min, 22 cycles of (98°C for 15 s, 67°C for 20 s, 72°C for 6 min), and final extension at 72°C for 5 min.

The amplified cDNA product was purified using VAHTS DNA Clean Beads. Sequencing libraries were prepared using TruePrep DNA Library Prep Kit V2 for Illumina (Vazyme, TD503-02).

Primer sequences: RT primer (Sangon), 5′Biotin-AAGCAGTGGTATCAACGCAGAGTACTTTTT TTTTTTTTTTTTTTTTTTTTTTTTTTTVN; TSO primer (Sangon), 5′Biotin-AAGCAGTGGTATCAAC CAGA GTACAT/rG//rG//iXNA_G/; ISPCR primer (Sangon), 5′Biotin-AAGCAGTGGTATCAACGC AGA*G*T.

## RNA-seq analysis

Raw fastq reads were trimmed by Cutadapt (version 1.18) to trim adapter and low-quality sequence.

Reads were aligned to the mouse genome (mm10) using STAR (version 2.5.3). The number of reads within each gene in each single cell was counted using RSEM (version 1.3.0) with gene annotation file from GENCODE (GRCm38.m23).

Differential expression was estimated by using DESeq2 package (version 1.26) with absolute log2 Fold change >0.5 and adjusted p-value <0.05.

Functional enrichment analysis was performed by using the clusterProfiler R package (v3.14.3) with default parameters (p-value <0.05) and the functional annotations terms in Gene Ontology (GO) (*Ashburner et al., 2000*).

## ATAC-sequencing

ATAC-seq were performed using TruePrep DNA Library Prep Kit V2 for Illumina according to the manufacturer's recommendation. Briefly, live cells were sorted using PI staining, then $5 \times 10^4$ cells were transferred to a low-binding centrifuge tube and resuspended in 50 µl of pre-chilled Lysis Buffer. After 20 times pipetting, cells were incubated on ice for 10 min for lysis. Subsequently, cells were pelleted by centrifugation at 4°C 500 × $g$ for 5 min, and suspended in a mixture consisting of 10 µl of 5× TTBL, 5 µl of TTE Mix V50, and 35 µl of ddH$_2$O. The mixture was gently mixed by pipetting 20 times, and incubated at 37°C for 30 min for fragmentation. The products were purified using 2× VAHTS DNA Clean Beads, eluted with 26 µl of water, and collected for the subsequent amplification. 24 µl of the eluted samples were amplified with different combination of i7 index and i5 index primers. QPCR were performed to determine the cycle number for PCR amplification.0.55× VAHTS DNA Clean Beads was employed for size selection of the PCR product. Finally, 30 ng of each sample were pooled for subsequent second-generation sequencing, on the Illumina NovaSeq 6000 with read length of 150 bp.

## ATAC-seq analysis

Raw fastq reads were trimmed by Cutadapt (version 1.18) to trim adapter and low-quality sequence.

Reads were aligned to the mouse genome (mm10) using Bowtie2 (version 2.3.3.1), the parameters are -t -q -N 1 -L 25 -X 2000 `--no-mixed --no-discordant`.

Reads with alignment quality <Q30, mapped to chrM, overlapping with ENCODE blacklisted regions were discarded. Duplicates were removed using Picard (version 2.20.4) and Samtools (version 1.13). Open chromatin region peaks were called on using MACS2 peak caller (version 2.2.7.1) with the following parameters: -nomodel -nolambda -call-summits. Peaks from all samples were merged and peaks falls in ENCODE blacklisted regions were filtered out.

Differential peaks were estimated by using 'edgeR' method in the R package DiffBind (version 2.14.0) with absolute log2 Fold change >1 and adjusted p-value <0.05.

The peaks annotations were performed with the 'annotatePeak' function in the R package ChIP-seeker (*Yu et al., 2015*).

The plot of ATAC-seq signals over a set of genomic regions were calculated by using 'computeMatrix' function in deepTools (version 2.0) and plotted by using 'plotHeatmap' and 'plotProfile' functions in deepTools.

## Cut&Tag assay

Foxp3-RFP+iTreg were induced, and infected with NFAT-Flag-GFP lentiviruses on days 1.5 and 2.5 of induction. On day 5, Foxp3-RFP+MSCV-NFAT-Flag-GFP+iTregs were sorted, then rested and restimulated for 2 days. Cut&Tag assay was performed using Hyperactive Universal CUT&Tag Assay Kit for Illumina (Vazyme Biotech Co., Ltd) according to the manufacturer's recommendation. Briefly, $1 \times 10^5$ iTregs were collected to low-binding centrifuge tubes and washed. Nucleus isolation were performed by NE buffer for 10 min on ice, and crosslinked by 0.1% formaldehyde for 2 min, then stopped by glycine. Con A Beads were pre-treated with binding buffer, and then incubated with nuclei of iTreg for 10 min. Further, samples were incubated with anti-Flag M2 antibody (1:50) or without primary antibody (Mock) overnight at 4°C. Goat Anti-Mouse IgG secondary antibody (1:50) was added and incubated for 1 hr. Cells were washed three times, and then incubated with pA/G-Tnp enzyme for 1 hr. After incubation, samples were mixed with 5× TTBL and Dig-300 Buffer and incubated at 37°C for 60 min for fragmentation. DNA was extracted and purified, then amplified with different combination of i7 index and i5 index primers. QPCR was performed to determine the cycle number for PCR amplification.VAHTS DNA Clean Beads were employed for size selection of the PCR product. Finally, 30 ng of each sample were pooled for subsequent second-generation sequencing, on the Illumina NovaSeq 6000 with read length of 150 bp.

## Cut&Tag data analysis

Raw fastq reads were trimmed by Cutadapt (version 1.18) to trim adapter and low-quality sequence.

Reads were aligned to the mouse genome (mm10) using Bowtie2 (version 2.3.3.1), the parameters are -t -q -N 1 -L 25 -X 2000 `--no-mixed --no-discordant`.

Reads with alignment quality <Q30, mapped to chrM, overlapping with ENCODE blacklisted regions (https://sites.google.com/site/anshulkundaje/projects/blacklists) were discarded. Duplicates were removed using Picard (version 2.20.4) and Samtools (version 1.13). Open chromatin region peaks were called using MACS2 peak caller (version 2.2.7.1) with the following parameters: -nomodel -nolambda -call-summits. Peaks from all samples were merged and peaks falls in ENCODE blacklisted regions were filtered out.

The peaks annotations were performed with the 'annotatePeak' function in the R package ChIP-seeker (*Yu et al., 2015*).

The plot of Cut&Tag signals over a set of genomic regions were calculated by using 'computeMatrix' function in deepTools and plotted by using 'plotHeatmap' and 'plotProfile' functions in deepTools.

The motif enrichment analysis was performed by using the 'findMotifsGenome.pl' command in HOMER with default parameters.

The motif occurrences in each peak were identified by using FIMO (MEME suite v5.0.4) with the following settings: a first-order Markov background model, a p-value cutoff of $10^{-4}$, and PWMs (Position weight matrix) from the mouse HOCOMOCO motif database (v11).

## Statistical analysis

Calculation and graphing were done with Prism (GraphPad). Data were presented as mean ± standard error of the mean, unless indicated otherwise. Student's *t*-test was used for comparing endpoint means of different groups. p-value <0.05 was considered significant. *$p < 0.05$; **$p < 0.01$; ***$p < 0.001$; ****$p < 0.0001$; N.S., not significant. Numbers of independently performed experiment repeats are shown as *N*, biological replicates of each experiment as *n*.

## Acknowledgements

We thank Dr. Hai Qi for providing Foxp3-GFP mice, platE cell line and guidance of calcium-related experimental design. We thank Dr. Saiyong Zhu for providing TetO-FUW-flag-Nfatc1 plasmid. We also thank Imaging Core Facility, Technology Center for Protein Sciences of Tsinghua University for assistance. Y.S. is supported by the joint Peking-Tsinghua Center for Life Sciences, the National Natural Science Foundation of China General Program (31370878), State Key Program (31630023), and Innovative Research Group Program (81621002), CIHR (PJT-156334 and PJT-166155) and NSERC (RGPIN/03748-2018).

## Additional information

### Competing interests

Xiaoyu Hu: Reviewing editor, eLife. The other authors declare that no competing interests exist.

### Funding

| Funder | Grant reference number | Author |
| --- | --- | --- |
| National Natural Science Foundation of China | 31370878 | Yan Shi |
| National Natural Science Foundation of China | 31630023 | Yan Shi |
| National Natural Science Foundation of China | 81621002 | Yan Shi |
| Canadian Institutes of Health Research | PJT-166155 | Yan Shi |

| Funder | Grant reference number | Author |
|---|---|---|
| Canadian Institutes of Health Research | PJT-156334 | Yan Shi |
| Natural Sciences and Engineering Research Council of Canada | RGPIN/03748-2018 | Yan Shi |

The funders had no role in study design, data collection, and interpretation, or the decision to submit the work for publication.

### Author contributions

Huiyun Lyu, Conceptualization, Resources, Data curation, Formal analysis, Validation, Investigation, Methodology, Writing – original draft, Project administration, Writing – review and editing; Guohua Yuan, Conceptualization, Resources, Data curation, Software, Formal analysis, Investigation, Visualization, Methodology, Writing – original draft, G.Y. performed RNA-seq experiment and all sequencing data analysis; Xinyi Liu, Formal analysis, Investigation, Visualization, X.L. performed AFM-related experiments; Xiaobo Wang, Resources, Investigation, X.W. provides suggestions to calcium-related expreiments and RNA-seq data analysis; Shuang Geng, Resources, Investigation, S.G. provides suggestions to mouse related experiments and RNA-seq data analysis; Tie Xia, Resources, Investigation, T.X. provided biophysical mechanism on cell adhesion; Xuyu Zhou, Conceptualization, Investigation, X.Z. provided critical review on Treg biology; Yinqing Li, Resources, Investigation, Y.L. provided critical technical support for sequencing experiment and data analysis; Xiaoyu Hu, Conceptualization, Investigation, Writing – review and editing, X.H. provided overall critical insight, reviewed and edited manuscript; Yan Shi, Conceptualization, Resources, Supervision, Funding acquisition, Visualization, Methodology, Writing – original draft, Project administration, Writing – review and editing

### Author ORCIDs

Xiaoyu Hu ⓘD https://orcid.org/0000-0002-4289-6998
Yan Shi ⓘD https://orcid.org/0000-0002-6715-7681

### Ethics

This study was performed in strict accordance with the recommendations in the Guide for the Care and Use of Laboratory Animals of the National Institutes of Health. All of the animals were handled according to approved Institutional Animal Care and Use Committee (IACUC) protocols of the Tsinghua University. The protocol was approved by the Committee on the Ethics of Animal Experiments of the Tsinghua University (22-SY6). All surgery was performed under sodium pentobarbital anesthesia, and every effort was made to minimize suffering.

Reviewer #1 (Public Review): https://doi.org/10.7554/eLife.88874.3.sa1
Reviewer #2 (Public Review): https://doi.org/10.7554/eLife.88874.3.sa2
Author Response https://doi.org/10.7554/eLife.88874.3.sa3

## Additional files

### Supplementary files
• MDAR checklist

### Data availability

RNA/ATAC/CUT&TAG associated sequencing data have been deposited in GEO under accession code GSE246431. All data excepted from the sequencing data, generated or analyzed during this study are included in the Source Data files.

The following dataset was generated:

| Author(s) | Year | Dataset title | Dataset URL | Database and Identifier |
|-----------|------|---------------|-------------|--------------------------|
| Lyu H, Yuan G, Shi Y | 2023 | Sustained store-operated calcium entry utilizing activated chromatin state leads to instability in iTregs | https://www.ncbi.nlm.nih.gov/geo/query/acc.cgi?acc=GSE246431 | NCBI Gene Expression Omnibus, GSE246431 |

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

# Appendix 1

**Appendix 1—key resources table**

| Reagent type (species) or resource | Designation | Source or reference | Identifiers | Additional information |
|---|---|---|---|---|
| Strain, strain background (*Mus musculus*) | C57BL6/J | Jackson Laboratory | Strain #:000664 from Jackson Laboratory | |
| Strain, strain background (*Mus musculus*) | Foxp3-GFP (B6.Cg-Foxp3tm2Tch/J) | Hai Qi of School of Medicine, Tsinghua University | Strain #:006772 from Jackson Laboratory | |
| Strain, strain background (*Mus musculus*) | Foxp3-RFP | Zhou Xuyu of Institute of Microbiology, Chinese Academy of Sciences | Foxp3-RFP | |
| Strain, strain background (*Mus musculus*) | CD45.1 | Jackson Laboratory | Strain #:002014 from Jackson Laboratory | |
| Strain, strain background (*Mus musculus*) | OT-II | Hai Qi of School of Medicine, Tsinghua University | Strain #:004194 from Jackson Laboratory | |
| Antibody | Ultra-LEAF Purified anti-mouse CD3ε Antibody, Armenian Hamster monoclonal | Biolegend | 100340 | 0.5 µg/ml |
| Antibody | Ultra-LEAF Purified anti-mouse CD28 Antibody, Syrian Hamster monoclonal | Biolegend | 102116 | 1 µg/ml |
| Antibody | NFATc1 Antibody (7A6), mouse monoclonal | Santa Cruz | sc-7294 | 1:2000 |
| Antibody | NFATc2 Antibody (4G6-G5), mouse monoclonal | Santa Cruz | sc-7296 | 1:2000 |
| Antibody | Lamin A/C (4C11) Mouse mAb | CST | 4777 | 1:2000 |
| Antibody | β-Actin (13E5) Rabbit mAb | CST | 4970 | 1:1000 |
| Antibody | GAPDH (D4C6R) Mouse mAb | CST | 97166 | 1:1000 |
| Antibody | Monoclonal ANTI-FLAG M2 antibody produced in mouse | Sigma | F1804 | 1:50 |
| Antibody | Goat Anti-Mouse IgG Secondary Antibody, monoclonal | Sino Biological | SSA021 | 1:50 |
| Antibody | Histone H3 (D2B12) XP Rabbit mAb (ChIP Formulated) | CST | 4620 | 1:50 |
| Antibody | Anti-Histone H3 (tri methyl K4) antibody - ChIP Grade, Rabbit polyclonal | Abcam | ab8580 | 1:50 |
| Antibody | Goat Anti-Rabbit IgG Secondary Antibody, monoclonal | Sino Biological | SSA018 | 1:50 |
| Antibody | Biotin anti-mouse CD3ε Antibody, Armenian Hamster monoclonal | Biolegend | 100303 | 0.5 µg/ml |
| Antibody | FOXP3 Monoclonal Antibody (150D/E4), PE, mouse monoclonal | eBioscience | 12-4774-42 | 1:500 |
| Antibody | FOXP3 Monoclonal Antibody (150D/E4), Alexa Fluor 488, mouse monoclonal | eBioscience | 53-4774-41 | 1:500 |
| Antibody | CD4 Monoclonal Antibody (GK1.5), APC, rat monoclonal | eBioscience | 17-0041-82 | 1:500 |
| Antibody | CD25 Monoclonal Antibody (PC61.5), PE, rat monoclonal | eBioscience | 12-0251-82 | 1:500 |
| Antibody | CD44 Monoclonal Antibody (IM7), APC, rat monoclonal | eBioscience | 17-0441-83 | 1:500 |
| Antibody | PE anti-mouse CD62L, rat monoclonal | Biolegend | 104408 | 1:500 |

*Appendix 1 Continued on next page*

*Appendix 1 Continued*

| Reagent type (species) or resource | Designation | Source or reference | Identifiers | Additional information |
|---|---|---|---|---|
| Peptide, recombinant protein | Recombinant Human TGF-beta 1 Protein | R&D | 240-B-010 | |
| Peptide, recombinant protein | Recombinant Human IL-2 | R&D | 202-IL | |
| Peptide, recombinant protein | Purified Streptavidin | Biolegend | 405150 | |
| Commercial assay, kit | LIVE/DEAD Fixable Aqua Dead Cell Stain Kit, for 405 nm excitation | Invitrogen | L34965 | |
| Commercial assay, kit | Propidium Iodide | Beyotime | C1062M-3 | |
| Commercial assay, kit | Mouse CD4+ T Cell Isolation Kit | Stem cell | 19852 | |
| Commercial assay, kit | Mouse CD25 Treg Cell positive selection Kit | Stem cell | 18782 | |
| Commercial assay, kit | Foxp3/Transcription Factor Staining Buffer Set | eBioscience | 00-5523-00 | |
| Commercial assay, kit | NE-PER Nuclear and Cytoplasmic Extraction Reagents | Thermo | 78835 | |
| Commercial assay, kit | TruePrep DNA Library Prep Kit V2 for Illumina | Vazyme | TD501 | |
| Commercial assay, kit | Hyperactive Universal CUT&Tag Assay Kit for Illumina | Vazyme | TD903 | |
| Commercial assay, kit | TruePrep Index Kit V2 for Illumina | Vazyme | TD202 | |
| Commercial assay, kit | VAHTS DNA Clean Beads | Vazyme | N411 | |
| Commercial assay, kit | Mouse IL-21 Uncoated ELISA | Invitrogen | 88–8210 | |
| Chemical compound, drug | Cyclosporin A | MCE | HY-B0579 | |
| Chemical compound, drug | CM-4620 | MCE | HY-101942 | |
| Chemical compound, drug | Thapsigargin | Invitrogen | T7459 | |
| Chemical compound, drug | Ionomycin | beyotime | S1672 | |
| Chemical compound, drug | Caffeine | Aladdin | C106953 | |
| Chemical compound, drug | 4-CMC | Sigma | C55402 | |
| Chemical compound, drug | Fluo-4, AM, cell permeant | Thermo | F14201 | |
| Chemical compound, drug | Cal RedTM R525/650 AM | AAT Bioquest | 20591 | |
| Chemical compound, drug | Indo-1 AM | BD | 565879 | |
| Chemical compound, drug | Pluronic F-127 | Sigma | P2443 | |
| Chemical compound, drug | CellTak 1 MG WI, 1/CS | BD | 354240 | |
| Chemical compound, drug | Hanks' Solution | Coolaber | SL6080 | |

*Appendix 1 Continued*

| Reagent type (species) or resource | Designation | Source or reference | Identifiers | Additional information |
|---|---|---|---|---|
| Chemical compound, drug | HBSS, 10× (without Calcium) | Macgene | CC016 | |
| Chemical compound, drug | Imject Alum | Thermo | 77161 | |
| Chemical compound, drug | OVA Peptide (323-339) | GenScript | RP10610-1 | |
| Chemical compound, drug | OVA | Sigma | A5503 | |
| Chemical compound, drug | Cell Trace CFSE | Thermo | C34554 | |
| Chemical compound, drug | Retinoic Acid | Sigma | R2625 | |
| Chemical compound, drug | InSolution Rapamycin | Sigma | 553211 | |
| Chemical compound, drug | L-Ascorbic acid (Vitamin C) | Sigma | A4403 | |
| Chemical compound, drug | AS2863619 | MCE | HY-126675A | |
| Chemical compound, drug | BAY 11-7082 | MCE | HY-13453 | |
| Chemical compound, drug | SP600125 | MCE | HY-12041 | |
| Chemical compound, drug | T-5224 | MCE | HY-12270 | |
| Software, algorithm | FlowJo | FlowJo LLC | | |
| Software, algorithm | Prism | GraphPad | | |
| Software, algorithm | IGV | Broad Institute | | |
| Software, algorithm | Imaris | Andor | | |
| Software, algorithm | R | R studio | | |
| Software, algorithm | JPK Data Processing | JPK | | |

