## [Editor Report · eLife assessment]

This study presents the **valuable** finding that sustained calcium signaling in induced-Treg (iTreg) cells can lead to the loss of Foxp3 expression and iTreg identity by altering the chromatin landscape. The evidence supporting the claims of the authors is **convincing**. The work will be of interest to immunologists working on Treg cell therapy.

---

## [Referee Report · Reviewer #1 (Public Review)]

This study revealed that one of the mechanisms for iTreg (induced-Treg) lineage instability upon restimulation is through sustained store-operated calcium entry (SOCE), which activates transcription factor NFAT and promotes changes in chromatin accessibility to activated T cell-related genes. The authors revealed that, unlike thymus-derived Tregs (tTreg) with blunted calcium signaling and NFAT activation, iTregs respond to TCR restimulation with fully activated SOCE and NFAT similar to activated conventional T cells. Activated NFAT binds to open chromatin regions in genes related to T helper cells, increases their expression, and leads to the instability of iTreg cells. On the other hand, inhibition of the SOCE/NFAT pathway by chemical inhibitors could partially rescue the loss of Foxp3 expression in iTreg upon restimulation. The conclusion of the study is unexpected since previous studies showed that NFAT is required for Foxp3 induction and iTreg differentiation (Tone Y et al, Nat Immunol. 2008, PMID: 18157133; Vaeth M et al, PNAS, 2012, PMID: 22991461). Additionally, Foxp3 interacts with NFAT to control Treg function (Wu Y et al, Cell, 2006, PMID: 16873067). The data presented in this study demonstrated the complex role NFAT plays in the generation and stability of iTreg cells.

Several concerns are raised from the current study.

1. Previous studies showed that iTregs generated in vitro from culturing naïve T cells with TGF-b are intrinsically unstable, and prone to losing Foxp3 expression due to lack of DNA demethylation in the enhancer region of the Foxp3 locus (Polansky JK et al, Eur J Immunol., 2008, PMID: 18493985). It is known that removing TGF-b from the culture media leads to rapid loss of Foxp3 expression. In the current study, TGF-b was not added to the media during iTreg restimulation, therefore, the primary cause for iTreg instability should be the lack of the positive signal provided by TGF-b. NFAT signal is secondary at best in this culturing condition.

This point has been addressed in the revision. Figure Q1 could be added to the manuscript as a supplementary figure.

2. It is not clear whether the NFAT pathway is unique in accelerating the loss of Foxp3 expression upon iTreg restimulation. It is also possible that enhancing T cell activation in general could promote iTreg instability. The authors could explore blocking T cell activation by inhibiting other critical pathways, such as NF-kb and c-Jun/c-Fos, to see if a similar effect could be achieved compared to CsA treatment.

This point has been sufficiently addressed in the revision.

3. The authors linked chromatin accessibility and increased expression of T helper cell genes to the loss of Foxp3 expression and iTreg instability. However, it is not clear how the former can lead to the latter. It is also not clear whether NFAT binds directly to the Foxp3 locus in the restimulated iTregs and inhibits Foxp3 expression.

This point has been addressed in the rebuttal. Could the authors incorporate their comments in the rebuttal into the discussion section of the revised manuscript?

---

## [Referee Report · Reviewer #2 (Public Review)]

The phenotypic instability of in vitro-induced Treg cells (iTregs) has been discussed for a long time, mainly in the context of the epigenetic landscape of Treg-signature genes; e.g. Treg-specifically CpG-hypomethylated Foxp3 CNS2 enhancer region. However, it has been insufficiently understood the upstream molecular mechanisms, the particularity of intracellular signaling of natural Treg cells, and how they connect to stable/unstable suppressive function.

Huiyun Lv et al. addressed the issue of phenotypic instability of in vitro-induced regulatory T cells (iTregs), which is a different point from the physiological natural Treg cells and an obstacle to the therapeutic use of iTreg cells. The authors focused on the difference between iTreg and nTreg cells from the perspective of their control of store-operated calcium entry (SOCE)-mediated cellular signaling, and they clearly showed that the sustained SOCE signaling in iTreg and nTreg cells led to phenotypic instability. Moreover, the authors pointed out the correlation between the incomplete conversion of chromatin configuration and the NFAT-mediated control of effector-type gene expression profile in iTreg cells. These findings potentially cultivate our understanding of the cellular identity of regulatory T cells and may shed light on the therapeutic use of Treg cells in many clinical contexts.

The authors demonstrated the biological contribution of Ca2+ signaling with the variable methods, which ensure the reliability of the results and the claims of the authors. iTreg cells sustained SOCE-signaling upon stimulation while natural Treg cells had lower strength and shorter duration of SOCE-signaling. The result was consistent with the previously proposed concept; a certain range of optimal strength and duration of TCR-signaling shape the Treg generation and maintenance, and it provides us with further in-depth mechanistic understanding.

In the later section, authors found the incomplete installment of Treg-type open chromatin landscape in some effector/helper function-related gene loci in iTreg cells. These findings propose the significance of focusing on not only the "Treg"-associated gene loci but also "Teffector-ness"-associated regions to determine the Treg conversion at the epigenetic level.

Limitations;

・NFAT regulation did not explain all of the differences between iTregs and nTregs, as the authors mentioned as a limitation.

・Also, it is still an open question whether NFAT can directly modulate the chromatin configuration on the effector-type gene loci, or whether NFAT exploits pre-existing open chromatin due to the incomplete conversion of Treg-type chromatin landscape in iTreg cells. The authors did not fully demonstrate that the distinct pattern of chromatin regional accessibility found in iTreg cells is the direct cause of an effector-type gene expression.

---

## [Author Response]

The following is the authors’ response to the original reviews.

**Reviewer #1 (Public Review):**
Several concerns are raised from the current study.(1) Previous studies showed that iTregs generated in vitro from culturing naïve T cells with TGF-b are intrinsically unstable and prone to losing Foxp3 expression due to lack of DNA demethylation in the enhancer region of the Foxp3 locus (Polansky JK et al, Eur J Immunol., 2008, PMID: 18493985). It is known that removing TGF-b from the culture media leads to rapid loss of Foxp3 expression. In the current study, TGF-b was not added to the media during iTreg restimulation, therefore, the primary cause for iTreg instability should be the lack of the positive signal provided by TGF-b. NFAT signal is secondary at best in this culturing condition.

In restimulation, void of TGFb is necessary to cause iTreg instability. Otherwise, the setup is similar to the iTreg-inducing environment (Author response image 1). On the other hand, the ultimate goal of this study is to provide a scenario that bears some resemblance of clinical treatment, where TGFb may not be available. The reviewer is correct in stating that TGFb is essential for iTreg stability, we are studying the role played by NFAT in iTreg instability in vitro, and possibly in potential clinical use of iTreg .

**Author response image 1. sa3fig1:** Restimulation with TGFb will persist iTreg inducing environment, resulting in less pronounced instability. Sorted Foxp3-GFP+ iTregs were rested for 1d, and then rested or restimulated in the presence of TGF-β for 2 d. Percentages of Foxp3+ cells were analyzed by intracellular staining of Foxp3 after 2 d.

(2) It is not clear whether the NFAT pathway is unique in accelerating the loss of Foxp3 expression upon iTreg restimulation. It is also possible that enhancing T cell activation in general could promote iTreg instability. The authors could explore blocking T cell activation by inhibiting other critical pathways, such as NF-kb and c-Jun/c-Fos, to see if a similar effect could be achieved compared to CsA treatment.

We thank the reviewer for this suggestion. We performed this experiment according to see extent of the role that NFAT plays, or whether other major pathways are involved. As Author response image 2 shows, solely inhibiting NFAT effectively rescued the instability of iTreg. The inhibition of NFkB (BAY 11-7082), c-Jun (SP600125), or a c-Jun/c-Fos complex (T5224) had no discernable effect, or in one case, possibly further reduction in stability.These results may indicate that NFAT plays a crucial and special role in TCR activation, which leads to iTreg instability. Other pathways, as far as how this experiment is designed, do not appear to be significantly involved.

**Author response image 2. sa3fig2:** Comparing effects of NFAT, NF-kB and c-Jun/c-Fos inhibitors on iTreg instability. Sorted Foxp3-GFP+ iTregs were rested for 1d, then restimulated by anti-CD3 and CD28 in the presence of listed inhibitors. Percentages of Foxp3+ cells were analyzed by intracellular staining after 2d restimulation.

(3) The authors linked chromatin accessibility and increased expression of T helper cell genes to the loss of Foxp3 expression and iTreg instability. However, it is not clear how the former can lead to the latter. It is also not clear whether NFAT binds directly to the Foxp3 locus in the restimulated iTregs and inhibits Foxp3 expression.

T helper gene activation is likely to cause instability in iTregs by secreting more inflammatory cytokines, as shown in Figure Q9, for example, IL-21 secretion. Further investigation is needed to understand how these genes contribute to Foxp3 gene instability exactly. With our limited insight, there may be two possibilities. 1. IL-21 directly affects Foxp3 through its impact on certain inflammation-related transcription factors (TFs). 2. There could be an indirect relationship where NFAT has a greater tendency to bind to those inflammatory TFs when iTreg instability appears, promoting the upregulation of these Th genes like in activated T cells, while being less likely to bind to SMAD and Foxp3, representing a competitive behavior. We at the moment cannot comprehend the intricacies that lead to the differential effects on T helper genes and Treg related genes.

With that said, we have previously attempted to explore the direct effect of NFAT on Foxp3 gene locus. Foxp3 transcription in iTregs primarily relies on histone modifications such as H3K4me3 (Tone et al., 2008; Lu et al., 2011) rather than DNA demethylation (Ohkura et al., 2012; Hilbrands et al., 2016). Previous studies have reported that NFAT and SMAD3 can together promote the histone acetylation of Foxp3 genes (Tone et al., 2008). In our previous set of experiments, we simultaneously obtained information of NFAT binding sites and H3K4me3. In Foxp3 locus, we observed a decreasing trend in NFAT binding to the CNS3 region of Foxp3 in restimulated iTregs compared to resting iTregs (Author response image 3). Additionally, the H3K4me3 modification in the CNS3 region of Foxp3 decreased upon iTreg restimulation, but inhibiting NFAT nuclear translocation with CsA could maintain this modification at its original level (Author response image 3).

**Author response image 3. sa3fig3:** The NFAT binding and histone modification on Foxp3 gene locus. Genome track visualization of NFAT binding profiles and H3K4me3 profiles in Foxp3 CNS3 locus in two batches of dataset.

Based on these preliminary explorations, it is concluded that NFAT can directly bind to the Foxp3 locus, and it appears that NFAT decreases upon restimulation, resulting in a decrease in H3K4me3, ultimately leading to the close association of NFAT and Foxp3 instability. However, due to limited sample replicates, these data need to be verified for more solid conclusions. We speculate that during the induction of iTregs, NFAT may recruit histone-modifying enzymes to open the Foxp3 CNS3 region, and this effect is synergistic with SMAD. When instability occurs upon restimulation, NFAT binding to Foxp3 weakens due to the absence of SMAD's assistance, subsequently reducing the recruitment of histone modifications enzyme and ultimately inhibiting Foxp3 transcription.

**Reviewer #2 (Public Review):**
(1) Some concerns about data processing and statistic analysis.The authors did not provide sufficient information on statistical data analysis; e.g. lack of detailed descriptions about-the precise numbers of technical/biological replicates of each experiment-the method of how the authors analyze data of multiple comparisons... Student t-test alone is generally insufficient to compare multiple groups; e.g. figure 1.These inappropriate data handlings are ruining the evidence level of the precious findings.

We thank the reviewer for pointing out this important aspect. In the figure legend, numbers of independently-performed experiment repeats are shown as N, biological replicates of each experiment as n.Student’s t test was used for comparing statistical significance between two groups. In this manuscript, all calculations of significant differences were based on comparisons between two groups. There were no multiple conditions compared simultaneously within a single group, and thus, no other calculation methods were used.

(2) Untransparent data production; e.g. the method of Motif enrichment analysis was not provided.Thus, we should wait for the author's correction to fully evaluate the significance and reliability of the present study.

Per this reviewer’s request, we have provided detailed descriptions of the data analysis for Fig 5, including both the method section and the Figure legend, as presented below:

“The peaks annotations were performed with the “annotatePeak” function in the R package ChIPseeker (Yu et al, 2015).

The plot of Cut&Tag signals over a set of genomic regions were calculated by using “computeMatrix” function in deepTools and plotted by using “plotHeatmap” and “plotProfile” functions in deepTools.The motif enrichment analysis was performed by using the "findMotifsGenome.pl" command in HOMER with default parameters.

The motif occurrences in each peak were identified by using FIMO (MEME suite v5.0.4) with the following settings: a first-order Markov background model, a P value cutoff of 10-4, and PWMs from the mouse HOCOMOCO motif database (v11).”

Additionally, we have also supplemented the method section with further details on the analysis of RNA-seq and ATAC-seq data.

(3) Lack of evidence in human cells. I wonder whether human PBMC-derived iTreg cells are similarly regulated.

This is a rather complicated issue, human T cells express FoxP3 upon TCR stimulation (PNAS, 103(17): 6659–6664), whose function is likely to protect T cells from activation induced cell death, and does not offer Treg like properties. In contrast in mice, FoxP3 can be used as an indicator of Treg. Currently, this is not a definitive marker for Treg in human, our FoxP3 based readouts do not apply. Nevertheless, we have now investigated whether inhibiting calcium signaling or NFAT could enhance the stability of human iTreg. As shown in Author response image 4, we found that the proportion of Foxp3-expressing cells did not show significant changes across the different conditions, while the MFI analysis revealed that CsA-treated iTreg exhibited higher Foxp3 expression levels compared to both restimulated iTreg and rest iTreg. However, CM4620 had no significant effect on Foxp3 stability, consistent with the observation of its limited efficacy in suppressing human iTreg long term activation. In summary, our results suggest that inhibiting NFAT signaling through CsA treatment can help maintain higher levels of Foxp3 expression in human iTreg.

**Author response image 4. sa3fig4:** Effect of inhibiting NFAT and calcium on human iTreg stability. Human naïve CD4 cells from PBMC were subjected to a two-week induction process to generate human iTreg. Subsequently, human iTreg were restimulated for 2 days with dynabeads followed by 2 days of rest in the prescence of CsA and CM-4620. Four days later, percentages of Foxp3+ cells and Foxp3 mean fluorescence intensity (MFI) were analyzed by intracellular staining.

(4) NFAT regulation did not explain all of the differences between iTregs and nTregs, as the authors mentioned as a limitation. Also, it is still an open question whether NFAT can directly modulate the chromatin configuration on the effector-type gene loci, or whether NFAT exploits pre-existing open chromatin due to the incomplete conversion of Treg-type chromatin landscape in iTreg cells. The authors did not fully demonstrate that the distinct pattern of chromatin regional accessibility found in iTreg cells is the direct cause of an effector-type gene expression.

To our surprise, the inhibition of NFkB (BAY 11-7082), c-Jun (SP600125), and the c-Jun/c-Fos complex (T5224) resulted in minimal alterations, as shown in Fig Q1. This seems to argue that NFAT may play a more special role in events leading iTreg instability.

We hypothesize that NFAT takes advantage of pre-existing open chromatin state due to the incomplete conversion of chromatin landscape in iTreg cells. Because iTreg cells, after induction, already exhibit inherent chromatin instability, with highly-open inflammatory genes. Furthermore, when iTreg cells were restimulated, the subsequent change in chromatin accessibility was relatively limited and not rescued by NFAT inhibitor treatment (Author response image 5). Therefore, in the case of iTreg cells, we propose that NFAT exploits the easy access of those inflammatory genes, leading to rapid destabilization of iTreg cells in the short term.

In contrast, tTreg cells possess a relatively stable chromatin structure in the beginning, it would be interesting to investigate whether NFAT or calcium signaling could disrupt chromatin accessibility during the activation or expansion of tTreg cells. It is possible that NFAT might cause the loss of the originally established demethylation map and open up inflammatory loci, thereby inducing a shift in gene transcriptional profiles, equally leading to instability.

**Author response image 5. sa3fig5:** Chromatin accessibility of Rest, Retimulated, CsA/ORAIinh treated restimulated iTreg. PCA visualization of chromatin accessibility profiles of different cell types. Color indicates cell type.

To establish a direct relationship between gene locus accessibility and its overexpression, a controlled experimental approach can be employed. One such method involves precise manipulation of the accessibility of a specific genomic locus using CRISPR-mediated epigenetic modifications at targeted loci. Subsequently, the impact of this manipulation on the expression level of the target gene can be precisely examined. By conducting these experiments, it will be possible to determine whether the augmented gene accessibility directly causes the observed gene overexpression.

**Reviewer #1 (Recommendations For The Authors):**
It might be helpful to add TGF-b to the iTreg restimulation culture to remove the influence of the lack of TGF-b from the equation, and measure the influence of SOCE/NFAT on iTreg instability.

Please refer to Author response image 1.

Alternatively, authors can also culture iTreg cells with TGF-b for 2 weeks when they undergo epigenetic changes and become more stabilized (Polansky JK et al, Eur J Immunol., 2008, PMID: 18493985). At this point, the stabilized iTregs can be used to measure the influence of SOCE/NFAT on iTreg instability.

In the study conducted by Polansky, it was observed in Figure 1 that prolonged exposure to TGF-β fails to induce stable Foxp3 expression and demethylation of the Treg-specific demethylated region (TSDR). Based on this finding, we could consider exploring alternative approaches to obtain a more stabilized iTreg population. One such approach could be isolating Foxp3+helios-Nrp1- iTreg cells directly from the peripheral in vivo, which are also known as pTregs. Generally, pTreg cells generated in vivo tend to be more stable compared to iTreg cells induced in vitro, and they already exhibit partial demethylation of the Treg signature, as shown in Fig 6C (Polansky JK et al, Eur J Immunol., 2008, PMID: 18493985). Investigating the role of NFAT and calcium signaling in pTreg cells would provide further insights into the additional roles of NFAT in Treg phenotypical transitions, particularly its role in chromatin accessibility.

In Figure 3, NFAT binding to the inflammatory genes in iTreg cells was even stronger than in activated T conventional cells. This is possibly due to Tconv cells being stimulated only once while iTregs were restimulated. A fair comparison should be conducted with restimulated activated conventional T cells.

Figure 3 demonstrates the accessibility of inflammatory gene loci, rather than NFAT binding. Comparing restimulated Tconvs with restimulated iTreg cells is indeed a valuable suggestion, as their activation state and polarization in iTreg directions could lead to distinct chromatin accessibility. Although one is activated long term regularly and the other is activated long term under iTreg polarization, it is highly likely that the chromatin state of both activated Tconvs and iTreg cells is highly open, especially in terms of the accessibility of inflammatory genes. This may provide us with a new perspective to understand iTreg cells, but will unlikely affect our central conclusion.

In the in vivo experiment in Figure 6, a control condition without OVA immunization should be included as a baseline.

We have performed this experiment in the absence of OVA, as depicted in Author response image 6. In the absence of OVA immunization, both WT-ORAI and DN-ORAI iTreg exhibited substantial stability, although DN-ORAI demonstrated a slightly less stable trend. Upon activation with 40ug and 100ug of OVA, DN-ORAI iTreg demonstrated enhanced stability than WT-ORAI iTreg, maintaining a higher proportion of Foxp3 expression.

**Author response image 6. sa3fig6:** Stability of DN-ORAI iTreg in vivo with or without OVA immunization. WT-ORAI/DN-ORAI-GFP+-transfected CD45.2+ Foxp3-RFP+ OT-II iTregs were transferred i.v. into CD45.1 mice. Recipients were left or immunized with OVA323-339 in Alum adjuvant. On day 5, mLN were harvested and analyzed for Foxp3 expression by intracellular staining.

**Reviewer #2 (Recommendations For The Authors):**
MajorSome concerns about the data processing and statistic analysis, as mentioned in the public review.In the figure legend, what does it mean e.g. n=3, N=3? Technical triplicate experiments? Three mice? Independently-performed three experiments? The authors should define it at least in the "Statistical analysis" in the method section otherwise the readers cannot determine the reason why they mainly use SEM for the data description.Moreover, in some cases, the number of experiments was not sure; e.g., Fig.1B, Fig. 5.How did the authors analyze data including multiple comparisons? Student t-test alone is generally insufficient to compare multiple groups; e.g. figure 1.

We thank the reviewer for pointing out this omission. Now, in the figure legend, numbers of independently-performed experiment repeats are shown as N, biological replicates of each experiment as n.For Fig. 1B, N=2, and for Fig 5, we have acquired NFAT Cut&Tag data for 2 times, N=2.Student’s t test was used for comparing statistical significance between two groups. In this manuscript, all calculations of significant differences were based on comparisons between two groups. There were no multiple conditions compared simultaneously within a single group, and thus, no other calculation methods were involved apart from the Student's t-test.

In Figure 1A, the difference in suppressiveness seemed subtle. Data collection of multiple doses of Tconv:Treg ratio will enhance the reliability of such kind of analysis.

We have now attempted the suppression assay with varying Treg:Tconv ratios and observed that the suppressive effect of iTreg was more obvious than that of tTreg when co-cultured at a 1:1 ratio with Tconv cells. However, as the cell number of tTreg and iTreg decreased, the inhibitory effects converged.

**Author response image 7. sa3fig7:** Compare multiple dose of Tconv:Treg ratio in suppression functionCFSE-labelled OT-II T cells were stimulated with OVA-pulsed DC, then different number of Foxp3-GFP+ iTregs and tTregs were added to the culture to suppress the OT-II proliferation. After 4 days, CFSE dilution were analyzed. Left, Representative histograms of CFSE in divided Tconvs. Right, graph for the percentage of divided Tconvs.

In Figure 3F, to which group did the shaded peaks belong? In this context, the authors should focus on "Activation Region" peaks (open chromatin signature in both TcAct & iTreg defined in Fig. 4D) but I did not find the peak in the focusing DNA regions in TcAct (e.g. the shaded regions in IL-4 loci). The clear attribution of the peaks to the heatmap will enhance the visibility and understanding of readers.

We have selected some typical peaks that belong to Fig 3D. These genes encompass some T-cell activation-associated transcription factors, such as Irf4, Atf3, as well as multiple members of the Tnf family including Lta, Tnfsf4, Tnfsf8, and Tnfsf14. Additionally, genes related to inflammation such as Il12rb2, Il9, and Gzmc are included. These genes show elevated accessibility upon T-cell activation, partially open in activated nTreg cells, referred to as the "Activation Region." They collectively exhibit high accessibility in iTreg cells, which may contribute to their instability.

**Author response image 8. sa3fig8:** Chromatin accessibility of some “Activation Region”. Genomic track showing chromatin accessibility of Irf4, Atf3, Lta, Tnfsf8, Tnfsf4, Tnsfsf14, Il12rb2, Il9, Gzmc in activated Tconv and iTreg.

In Figure 4A/S4A, the information on cell death will help the understanding of readers because the sustained SOCE is associated with cell survival as shown in Fig. S2. The authors can discuss the relationships between cell death and Foxp3 retention, which potentially leads to a further interesting question; e.g. the selective/resistance to activation-induced cell death as the identity of Treg cells.

As shown in Author response image 9, activated iTreg cells indeed exhibit a certain degree of cell death compared to resting iTreg cells. The inhibition of NFAT by CsA enhances the survival rate of iTreg cells, but the inhibition of ORAI by CM-4620 leads to more severe cell death. The cell death induced by CsA and CM-4620 is not consistent, indicating that there may not be a direct proportional relationship between cell death and the expression of Foxp3 and Treg identity.

**Author response image 9. sa3fig9:** Relationship of cell death and Foxp3 stability in restimulated iTregs. Sorted Foxp3-GFP+ iTregs were rested for 1d, then restimulated by anti-CD3 and CD28 in the presence of CsA or CM-4620. After 2d restimulation, live cell percentage were analyzed by staining of Live/Dead fixable Aqua, and percentages of Foxp3+ cells were analyzed by intracellular staining of Foxp3. Upper, live cell percentage of iTregs. Lower, percentages of Foxp3 in iTregs.

In Figure 5, the information for the data interpretation was insufficient.

We have provided detailed descriptions of the data analysis for Fig 5, including both the method section and the Figure legend, as presented below:

“The peaks annotations were performed with the “annotatePeak” function in the R package ChIPseeker (Yu et al, 2015).The plot of Cut&Tag signals over a set of genomic regions were calculated by using “computeMatrix” function in deepTools and plotted by using “plotHeatmap” and “plotProfile” functions in deepTools.The motif enrichment analysis was performed by using the "findMotifsGenome.pl" command in HOMER with default parameters.The motif occurrences in each peak were identified by using FIMO (MEME suite v5.0.4) with the following settings: a first-order Markov background model, a P value cutoff of 10-4, and PWMs from the mouse HOCOMOCO motif database (v11).”

Additionally, we have also supplemented the method section with further details on the analysis of RNA-seq and ATAC-seq data.

The correlation between the open chromatin status of the gene loci described in Fig.5E and the expression at mRNA level? e.g.; Do iTreg-Act cells produce a higher level of IL-21 than nTreg-act?The analysis in Fig.5F-G should be performed in parallel with nTreg cells to emphasize the distinct NFAT-chromatin regulation in iTreg cells.

We have now compared the secretion levels of IL-21 in tTreg and iTreg upon activation and treated with CsA by ELISA. As shown in Author response image 10, tTreg did not secrete IL-21 regardless of activation status (undetectable), while iTreg did not secrete IL-21 at resting state but exhibited IL-21 secretion after 48 h of activation. Moreover, the secretion of IL-21 was inhibited by CsA and CM-4620 treatment. This observation aligns with our earlier findings where we observed nuclear binding of NFAT to gene loci of these cytokines, enhancing their expression and pushing iTreg unstable under inflammatory conditions. These findings further underscore the likelihood that the inhibition of calcium and NFAT signaling might contribute to the stabilization of iTreg by suppressing the secretion of inflammatory cytokines.

**Author response image 10. sa3fig10:** IL-21 secretion in tTreg and iTreg upon activation. iTregs and tTregs were sorted and restimulated with anti-CD3 and anti-CD28 antibodies, in the presence of CsA and CM-4620. Cell culture supernatant were harvested after 2 d restimulation and IL-21 secretion was analyzed by ELISA.

Performing a parallel comparison of NFAT activity between tTreg and iTreg cells was initially part of our experimental plan. However, it proved challenging in practice, as we encountered difficulties in efficiently infecting tTreg cells with NFAT-flag. Consequently, we could not obtain a sufficient number of tTreg cells for conducting Cut&Tag experiments.

Based on our observations, we speculate that there might be substantial differences in the accessibility of genes in tTreg cells, leading to considerable variations in the repertoire of genes available for NFAT to regulate. As a result, we expect significant differences in the nuclear localization and activity of NFAT between iTreg and tTreg cells.

In Figure 6C, what does the FCM plot between Foxp3-CFSE look like?The authors can discuss the mechanism of ORAI-DN-mediated through such analysis; e.g. the possibility that selective proliferation defect by ORAI-DN in Foxp3- cells led to an increased percentage of Foxp3, not only just unstable transcription of Foxp3.

This is an in vitro experiment to assess the suppressive effect of iTreg on Tconv proliferation. Therefore, CFSE is used to stain Tconv cells, but not iTreg cells, so we did not detect proliferation feature of iTreg.

MinorConfusing terminology of "tTreg" at line 47, etc. "natural Treg" contains both thymic-derived Treg and periphery-derived Treg cells. (A Abbas et al. Nat Immunol. 2013)

We have now changed the designation to tTreg at line 47. tTreg refers to thymus-derived regulatory T cells, while nTreg includes both tTreg and pTreg. However, it is important to note that the Treg cells used in our study were isolated from the spleen of 2-4-month-old Foxp3-GFP or Foxp3-RFP mice. The CD4+ T cells were first enriched using the CD4 Isolation kit, and the FACSAriaII was utilized to collect CD4+ Foxp3-GFP/RFP+ Treg cells. Subsequently, Helios and Nrp-1 staining revealed that the majority of these cells were nTreg, with only approximately 6% being pTreg. Overall, we consider the cells we used as tTreg.

In all FCM analyses, the authors should clarify how to detect Foxp3 expression; Foxp3-GFP/Foxp3-RFP/Intracellular staining like Figure S5A (but not specified in the other FCM plots)

All Foxp3 expressions in the article were assessed using intracellular staining, as described in the methods section, and we have added specific descriptions to each figure legend. The reason for employing intracellular staining is that we used Foxp3-IRES-GFP mice, where GFP and Foxp3 are not fused into a single protein, existing as separate proteins after expression. Therefore, during induction, the appearance of GFP protein might potentially represent the presence of Foxp3. However, in cases of Foxp3 instability, the degradation of GFP protein may not be entirely synchronized with that of Foxp3 protein, making GFP an unreliable indicator of Foxp3 expression levels. As a result, for the purification of pure iTreg cells, we used Foxp3-GFP/RFP fluorescence, while for observing instability, we employed intranuclear staining of Foxp3.

In Figure 6B, the captions were lacking in the two graphs on the right side

The two restimulation conditions, 0.125+0.25 and 0.25+0.5, have been added into Fig 6B right side.

In Figure S2, the annotation of the x-y axis was missing.

Added.

Lack of reference at line 292.

Reference 42-46 were added.

In the method section, the authors should note the further product information of antibodies and reagents to enhance reproducibility and transparency. Making a list that clarifies the suppliers, Ab clone, product IDs, etc. is encouraged. The authors did not specify the supplier of recombinant proteins and which type of TGF-beta (TGF-beta 1, 2, or 3?).

A detailed description of the mice, antibodies, Peptide recombinant protein, commercial kit, and software has been provided and incorporated into the methods section.

In the method section, the authors should clarify which Foxp3-reporter strain. There are many strains of Foxp3-reporter mice in the world. In line 373, is the "FoxP3-IRES-GFP transgenic mice" true? Knock-in strain or BAC-transgene?

This mouse is a gift from Hai Qi Lab in Tsinghua University. They acquired this mouse strain from Jackson Laboratory, and the strain name is B6.Cg-Foxp3tm2Tch/J, Strain #:006772. An IRES-EGFP-SV40 poly A sequence was inserted immediately downstream of the endogenous Foxp3 translational stop codon, but upstream of the endogenous polyA signal, generating a bicistronic locus encoding both Foxp3 and EGFP.

The age of mice used in the experiments should be specified, and confusing words such as "young" should not be used in any method descriptions; e.g. line 405.

The detailed mouse age has been added in the methods section. “To prepare Tconv, tTreg and iTreg for experiments, spleen was isolated from 2-4-month-old Foxp3-GFP mice for Tconv and tTreg sorting, and 6-week-old mice for iTreg induction.”

The method of how the original ATAC-seq/Cut & Tag data were generated was not described in the method section.

Added in method section.

The reference section was incomplete, and the style was not unified. e.g.; ref 7, 24, 25, 26 ... I gave up checking all.

The style of ref 7, 22, 24, 26, 28, 31, 33, 35 were modified.

Changes in manuscript:

Author Name: “Huiyun Lv” to “Huiyun Lyu”.

Fig 1A was updated according to Reviwer 2’s suggestion.

Fig S3E and associated description was added according to Reviwer 2’s suggestion.

Fig S4C and associated description was added according to Reviwer 1’s suggestion.

Fig 5H and associated description was added according to Reviwer 2’s suggestion.

Fig 6D were updated according to Reviwer 1’s suggestion.

Fig 2D was corrected, the labels for gapdh and actin in the iTreg panel were inadvertently switched. The mistakehas been rectified, and the original gel image will be provided.

Fig 2A and Fig 4A was updated.

The style of Fig 6B and Fig S2A was modified.

Method:

Mice: FoxP3-IRES-GFP with more description.

Flow Cytometry sorting and FACS: the detailed mouse age has been added.RNA-seq analysis, ATAC-sequencing, ATAC-seq analysis, Cut&Tag assay, Cut&Tag data analysis: more description was added.

Statistical analysis: “Numbers of independently-performed experiment repeats are shown as N, biological replicates of each experiment as n.” were added.

Reference: Ref 42-46 and 49-52 were added. The style of ref 7, 22, 24, 26, 28, 31, 33, 35 were corrected.

A detailed description of the mice, antibodies, Peptide recombinant protein, commercial kit, and software has been provided.